# Internal tides vertical structure and steric sea surface height signature south of New Caledonia revealed by glider observations

Arne Bendinger[1], Sophie Cravatte[1,2], Lionel Gourdeau[1], Luc Rainville[3], Clément Vic[4], Guillaume Sérazin[4,a], Fabien Durand[1], Frédéric Marin[1], and Jean-Luc Fuda[5]

[1]Université de Toulouse, LEGOS (CNES/CNRS/IRD/UT3), Toulouse, France
[2]IRD, Centre IRD de Nouméa, New Caledonia
[3]Applied Physics Laboratory, University of Washington, Seattle, WA, USA
[4]Laboratoire d'Océanographie Physique et Spatiale, Univ. Brest, CNRS, Ifremer, IRD, IUEM, Brest, France
[5]Aix Marseille Univ., Université de Toulon, CNRS, IRD, MIO UM 110, Marseille, France
[a]now at: Institut de Recherche de l'Ecole Navale (IRENav), EA 3634 - Ecole Navale, 29240, Brest, France

**Correspondence:** Arne Bendinger (arne.bendinger@univ-tlse3.fr)

**Abstract.** In this study, we exploit autonomous underwater glider data to infer internal tide dynamics south of New Caledonia, an internal-tide generation hot spot in the southwestern tropical Pacific. By fitting a sinusoidal function to vertical displacements at each depth using a least-squares method, we simultaneously estimate diurnal and semidiurnal tides. Our analysis reveals regions of enhanced tidal activity, strongly dominated by the semidiurnal tide. To validate our findings, we compare the glider observations to a regional numerical simulation that includes tidal forcing. This comparison assesses the simulation's realism in representing tidal dynamics and evaluates the glider's ability to infer internal tide signals and their signature in sea surface height (SSH). The glider observations and a pseudo glider, simulated using hourly numerical model output with identical sampling, exhibit similar amplitude and phase characteristics along the glider track. Existing discrepancies are primarily explained by tidal incoherence induced by eddy-internal tide interactions. We infer the semidiurnal internal tide signature in steric SSH by the integration of vertical displacements. Within the upper 1000 m, the pseudo glider captures roughly 78 % of the steric SSH total variance explained by the full water column signal. This value increases to over 90 % when projecting the pseudo glider's vertical displacements onto climatological baroclinic modes and extrapolating to full depth. Notably, the steric SSH from glider observations aligns closely with empirical estimates derived from satellite altimetry, highlighting the glider observations' predominating coherent nature.

## 1 Introduction

Over the last two decades, in-situ observations (e.g., Park and Watts, 2006; Zilberman et al., 2011; Nash et al., 2012; Vic et al., 2018), satellite altimetry (Ray and Zaron, 2016; Zhao et al., 2016; Zaron, 2019), and numerical modeling (for a review see Arbic et al., 2018; Arbic, 2022) have shed light on internal-tide dynamics at both regional and global scales. Important internal

tide generation sites have been identified in regions such as the Hawaiian Islands, the Luzon Strait, the Indonesian Seas, French Polynesia, the southwestern tropical Pacific, Madagascar, the Amazonian shelf break, and the Mid-Atlantic Ridge. At these locations, the barotropic tidal flow interacts with the bathymetry while radiating internal waves at tidal frequency into the stably stratified water column, expressed by vertical displacements of density surfaces (Bell Jr, 1975; Baines, 1982).

Each of the above tools, namely in-situ observations, satellite altimetry, and numerical modeling, possesses its own set of benefits and limitations. In-situ observations such as moorings provide excellent temporal resolution and in most cases a sufficient vertical resolution to resolve the wave's vertical structure. However, these scattered in-situ measurements are only representative at very local scales. Satellite altimetry provides a global view of internal tides, but long time series are needed and the derived signal is mostly representative of low-vertical mode dynamics at large horizontal scales. Numerical modeling overcomes both of these issues by investigating the fine spatial and temporal scales over a large region. Though, high-resolution grid spacing is needed making numerical modeling computationally expensive. Further, these models and the underlying primitive equations may be simplifications of the complex reality that do not fully encapsulate the intricacies of the actual physical system. This concerns sub-grid scale physics such as unresolved dissipative effects which require parameterization through turbulent closure schemes. Generally, any interpretation and conclusion drawn from the above approaches should be taken with thoughtful consideration.

Among the in-situ platforms, gliders have the potential to infer the vertical structure of internal tides, while documenting their spatial variability. Traditionally used to document lower frequency features such as mesoscale and submesoscale features at high spatial resolution (Rudnick, 2016; Testor et al., 2019), they have the potential to complement knowledge obtained from moorings and satellite altimetry. Commonly, gliders are programmed to provide subsurface observations by sampling the upper ocean in a saw-tooth manner. For a maximum depth of 1000 m, a typical glider dive cycle is 6 h during which it travels 6 km horizontally. As they travel autonomously through the ocean over thousands of km and for a duration of the order of months per mission, they can sample a large area. This makes gliders advantageous compared to other in-situ platforms such as moorings, which are confined to fixed locations. However, glider measurements merge spatial and temporal variability and, thus, make it difficult to separate high-frequency signals from low-frequency (but spatially varying) motions such as mesoscale and submesoscale features.

Despite these limitations, glider measurements have been previously successfully exploited to access hydrographic data at fine-scale resolution, and to infer internal-tide dynamics in dedicated areas. Pioneer work using glider data was carried out by Rainville et al. (2013) and Johnston et al. (2013) who estimated amplitude and phase of diurnal and semidiurnal internal tides. Glider data were shown to capture the phase propagation away from the generation site and map the mode-1 energy flux at the Luzon Strait. Johnston and Rudnick (2015) extracted diurnal and semidiurnal internal tides from repeated glider cross-shore transects in the California Current System. They link the internal tide induced mixing with elevated diffusivity estimates. Johnston et al. (2015) revealed standing wave patterns in the Tasman Sea as incident mode-1 internal tides reflected

on the continental slope. Moreover, internal tides were extracted from gliders that were employed for vertical profiling, serving as fixed-point time series while maintaining station (Hall et al., 2017, 2019).

This study focuses on internal tides south of New Caledonia, an internal tide generation hot spot in the southwestern tropical Pacific (Fig. 1), recently described and quantified in Bendinger et al. (2023) using numerical modeling. Internal tide generation was found to be closely linked with the north-south stretching ridge system composed of shelf breaks, oceanic ridges, and seamounts which represent a major obstacle for the barotropic tidal flow bending around New Caledonia. In the full-regional domain, a total of 15.27 GW is converted from the barotropic to the baroclinic M2 tide, comparable to well-known sites of enhanced energy conversion (e.g. the Hawaiian Ridge). Barotropic-to-baroclinic energy conversion is associated with the main bathymetric structures, namely Grand Passage, Pines Ridge, Norfolk Ridge, and Loyalty Ridge (see Fig. 1), governed by the semidiurnal M2 tide and strongly dominated by mode 1. Tidal energy propagation is characterized by well-confined tidal beams that diverge away from the generation hot spots north and south of New Caledonia with depth-integrated energy fluxes of up to 30 kW m$^{-1}$ in the annual mean. The modeled internal-tide dynamics have been validated in terms of barotropic-to-baroclinic energy conversion against semianalytical theory. In addition, the surface signature has been validated against satellite altimetry products revealing reasonable amplitude and large-scale (interference) patterns. The above model analysis only concerned the coherent tide, which is the stationary component being constant in amplitude and phase. The departure from tidal coherence is referred to as tidal incoherence, i.e. the temporally varying amplitude and phase within the tidal frequency band. It is characterized by its unpredictability often linked to mesoscale variability and stratification changes both close to the generation site and during tidal energy propagation. Particularly, mesoscale turbulence and background currents were shown to cause a tidal beam refraction associated with changing phase speeds and, consequently, alterations in the propagation of internal tides (Rainville

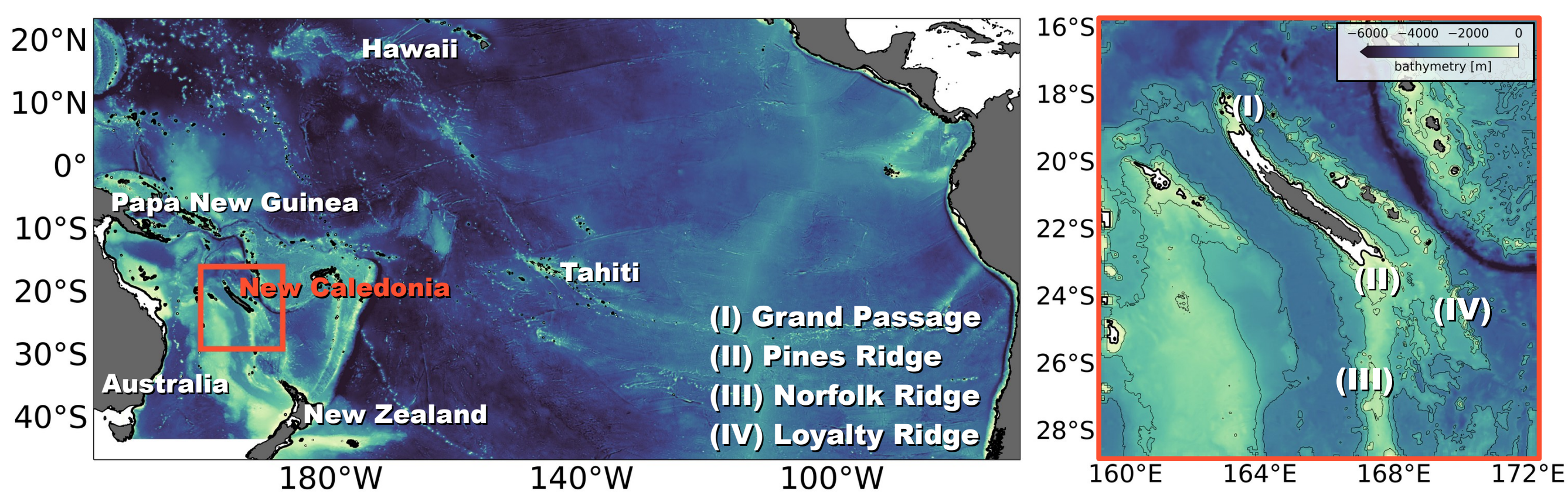

**Figure 1.** New Caledonia is located in the southwestern tropical Pacific, an area of complex bathymetry with continental shelves, shelf breaks, large- and small-scale ridges, and seamounts. It is subject to strong internal tide generation associated with the major bathymetric features, i.e. (I) Grand Passage, (II) Pines Ridge, (III) Norfolk Ridge, and (IV) Loyalty Ridge.

and Pinkel, 2006; Duda et al., 2018; Guo et al., 2023).

From an in-situ perspective, the numerical model results remain to be fully validated. In-situ observations of fine-scale physics in the region are rare. Moored measurements of velocity were used to compare kinetic energy frequency spectra with the numerical model output (Durand et al., 2017; Bendinger et al., 2023). However, the mooring is neither located close to a pronounced internal tide generation site, nor in propagation direction. Insight into fine-scale dynamics around New Caledonia is given by a unique set of glider surveys undertaken in the period from 2011 to 2014 (Durand et al., 2017). One of these glider missions surveyed the region of high internal tide activity south of New Caledonia, which is also a region of high mesoscale eddy activity (Keppler et al., 2018) and submesoscale activity (Sérazin et al., 2020). Disentangling balanced from unbalanced motions (mesoscale and submesoscale features from internal waves) in this area is a challenge of particular interest in the context of the Surface Water Ocean Topography (SWOT) satellite altimetry mission and the SWOT Adopt-A-Crossover (AdAC) initiative (d'Ovidio et al., 2019; Morrow et al., 2019). SWOT will provide high-resolution sea surface height (SSH) measurements along two swaths of 60 km width each resolving wavelengths down to 15 km, which is up to ten times higher resolution than conventional altimetry (Fu et al., 2012; Ballarotta et al., 2019; d'Ovidio et al., 2019; Morrow et al., 2019). The availability of three-dimensional in-situ observations may provide insight into the SSH expression of fine-scale dynamics, with important implications for disentangling SWOT SSH measurements. New Caledonia represents an interesting site for addressing mesoscale and submesoscale SSH observability in a region with strong internal tides. Specifically, the glider data can be very useful in linking the vertical structure of the ocean interior with the ocean surface. Although not suitable for the direct assessment of SWOT, it represents an important in-situ dataset with relevant information about the governing dynamics at play.

This study's objective lies in the exploitation of the glider's spatio-temporal sampling in the upper 1000 m to infer internal-tide dynamics, including their steric SSH signature south of New Caledonia. To assess our findings, we seek a complementary validation of the regional numerical simulation by glider observations and vice-versa. On the one hand, the glider observations will assess the realism of internal tides' simulation in the regional model. On the other hand, the regional model will address the capability of the methodologies applied to the glider observations. Specifically, we address the following questions: How do the observation-based and simulated internal tide fields compare with each other? Can observations and the model be used complementarily to deduce tidal coherence and/or tidal incoherence? To what extent are the glider observations of the upper 1000 m sufficient to deduce the internal tide steric SSH signature?

## 2 Data

### 2.1 Glider observations

This study focuses on autonomous Spray glider observations obtained during a mission from 12 August 2014 to 23 October 2014 within a series of glider surveys around New Caledonia in the framework of the Southwest Pacific Ocean Circulation and Climate Experiment (SPICE, Ganachaud et al., 2014; Durand et al., 2017). The glider surveyed continuously temperature

and salinity with respect to pressure in the upper 1000 m as it travelled horizontally and vertically in the water column while sending its GPS location before each descending and after each ascending profile (Fig. 2).

Along the glider's path, a total of 560 (ascending/descending) profiles were analyzed which were acquired over the course of 73 days and a horizontal travel distance of 1150 km. The glider track as a function of days since deployment is shown in Fig. 3. The glider was deployed at the southern edge of the New Caledonia lagoon at 166° E, 22° S before heading south to 26.5° S and heading back north to its initial starting position. The mean duration of a glider profile is 2.9 h (3.4 h for the mean descending profile, 2.4 h for the mean ascending profile). The mean horizontal displacement is 2.1 km (2.4 km for the mean

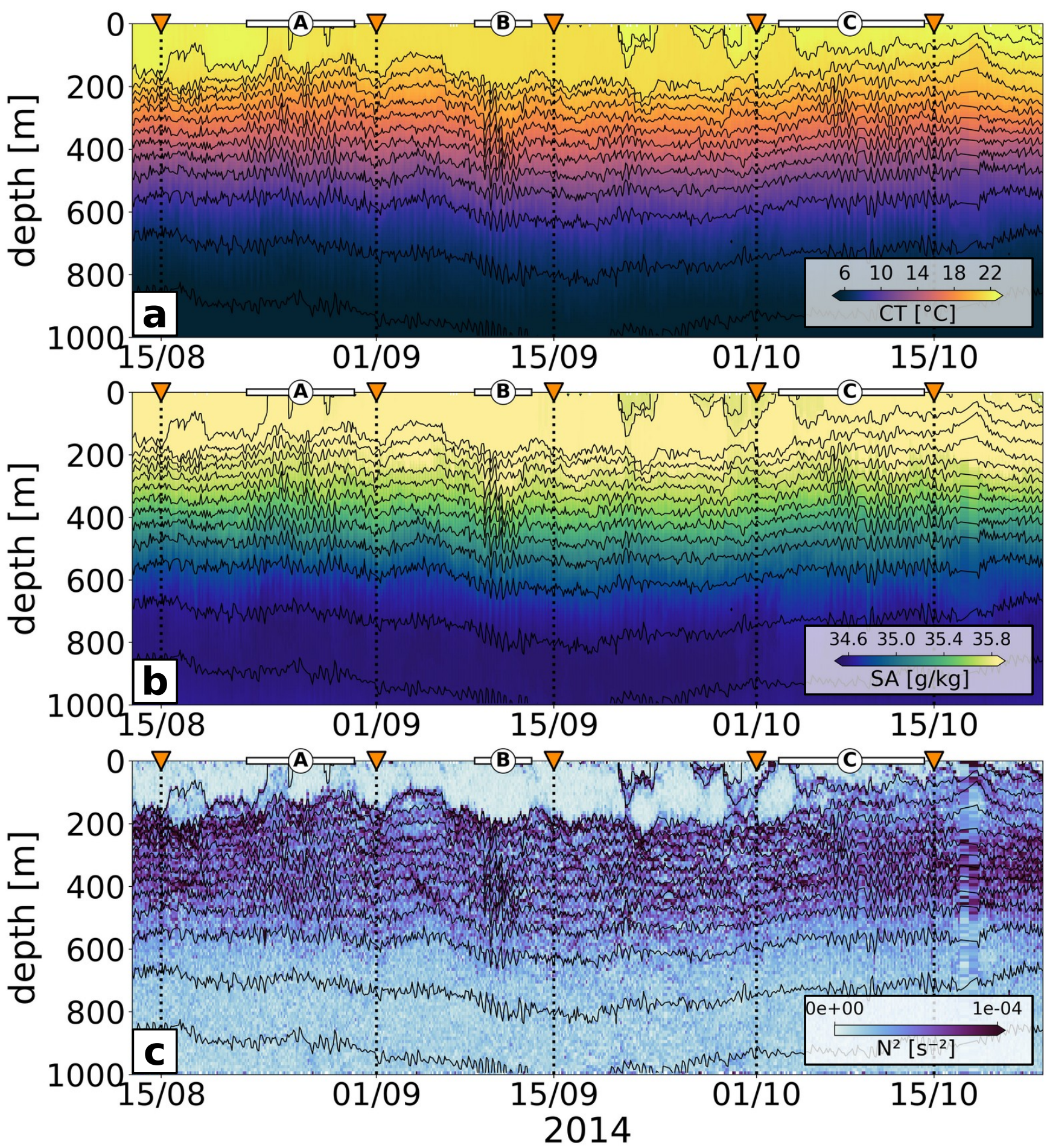

**Figure 2.** Glider observations (a) conservative temperature (CT), (b) absolute salinity (SA), and (c) squared buoyancy frequency ($N^2$) overlaid by potential density contours and gridded in 10 m bins along the vertical axis. The orange triangles (and the dotted vertical lines) mark the glider way points as given in Fig. 3. The white bars represent the sections of potential tidal beam crossing, namely A, B, C (as defined in Fig. 3).

descending profile, 1.7 km for the mean ascending profile). Aborted glider profiles as well as profiles featuring faulty GPS data were discarded from the analysis. The acquired glider time series of temperature and salinity with respect to pressure were divided into descending and ascending profiles by allocating the glider time stamp with the maximum dive depth. The profiles were then vertically gridded and binned in 10 m depth intervals.

## 2.2 Numerical simulation

This study uses numerical output of a model configuration that consists of a host grid (TROPICO12, 1/12°horizontal resolution and 125 vertical levels) and covers the tropical and subtropical Pacific Ocean basin from 142° E-70° W and 46° S-24° N (Fig. 1), as introduced in Bendinger et al. (2023). The oceanic reanalysis GLORYS2V4 prescribes initial conditions for temperature and salinity as well as the forcing with daily currents, temperature, and salinity at the open lateral boundaries. ERA5 produced by the European Centre for Medium-Range Weather Forecasts (ECMWF, Hersbach et al., 2020) provides atmospheric forcing at hourly temporal resolution and a spatial resolution of 1/4°to compute surface fluxes using bulk formulae and the

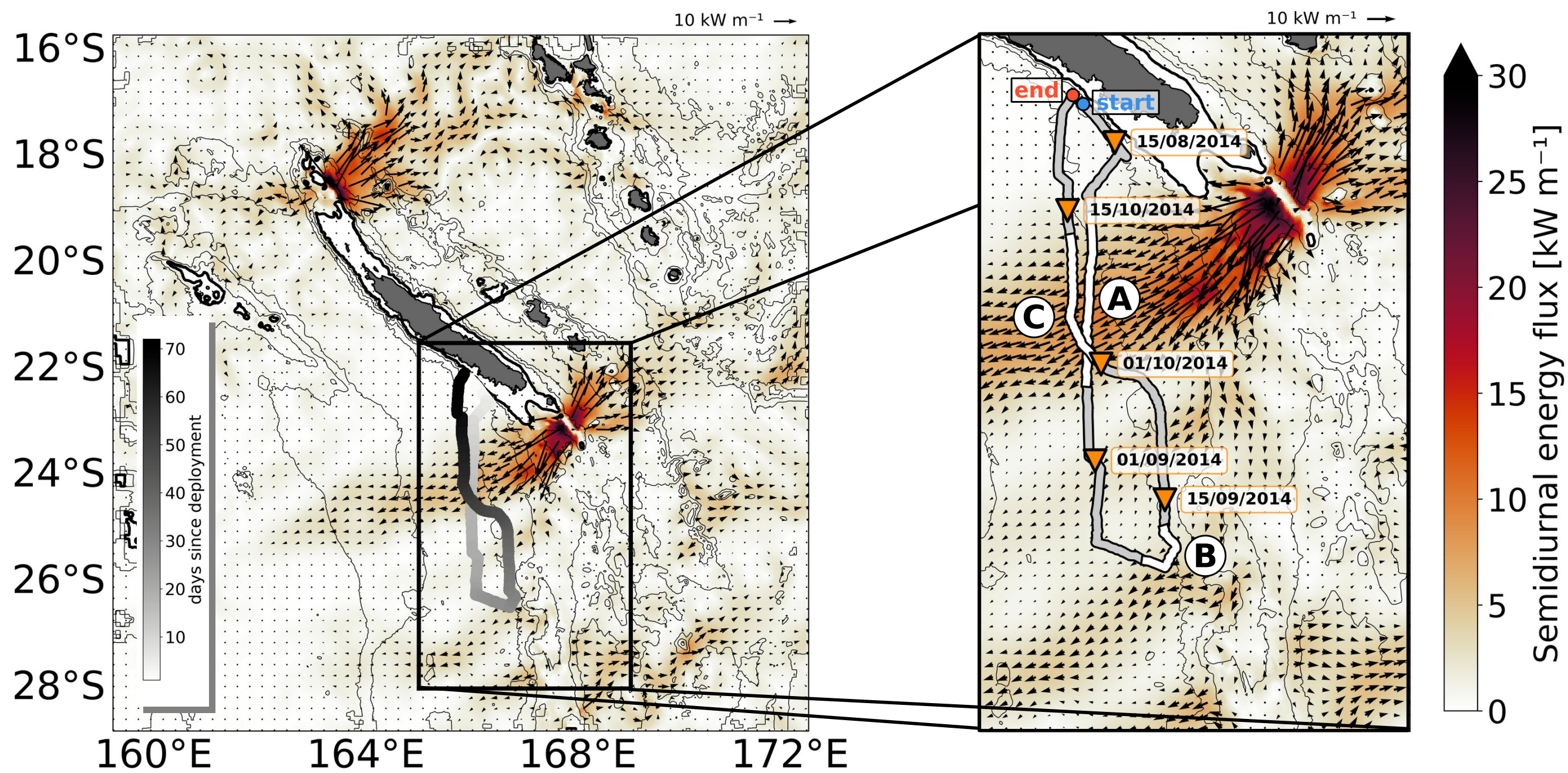

**Figure 3.** CALEDO60 regional model domain showing the depth-integrated coherent semidiurnal energy flux (shading) including flux vectors and the glider track of the glider mission south of New Caledonia as a function of days since deployment. The inset zooms into the study area. For the sake of better visualization and orientation, orange triangles and the associated time stamps mark way points along the glider track (15 August, 1 September, 15 September, 1 October, and 15 October 2014. The highlighted white segments (A, B, C) illustrate potential major crossings of glider track and tidal beam energy propagation. The thin black lines represent the 1000, 2000, and 3000 m depth contours. The thick black line is the 100 m depth contour representative for the New Caledonia lagoon.

model prognostic sea surface temperature. The model is forced by the tidal potential of the five major diurnal (K1, O1) and semidiurnal (M2, S2, N2) tidal constituents. At the open lateral boundaries it is forced by barotropic sea surface height and barotropic currents of the same five tidal constituents taken from the global tide atlas FES2014 (Finite Element Solution 2014, Lyard et al., 2021). A higher-resolution horizontal grid is nested within the host grid in the southwestern tropical Pacific Ocean encompassing New Caledonia (Fig. 3). This nesting grid features 1/60°horizontal resolution or ∼1.7 km grid box spacing initialized by an Adaptive Grid Refinement in Fortran (AGRIF, Debreu et al., 2008). AGRIF was explicitly designed for NEMO to set up regional simulations embedded in a pre-defined model configuration. Further, it enables the two-way lateral boundary coupling between the host and the nesting grid during the whole length of the simulation. Bendinger et al. (2023) illustrated the model's eligibility of realistically simulating both background ocean dynamics (i.e. large-scale circulation, kinetic energy spectra) and tidal dynamics. We refer to Bendinger et al. (2023) for a more detailed model description and model assessment.

Here, we focus on the model year 2014 using the hourly regional model output (CALEDO60) of the three-dimensional velocity field, temperature, salinity, pressure. This model output was also subject to a coherent tidal analysis at each grid point providing the tidal harmonics for the diurnal and semidiurnal constituents as well as tidal energetics. The harmonic analysis constraint to a full calendar year relies upon a compromise between high computational expenses and the representative extraction of the coherent tide through a time series long enough to sample a representative variability of the mesoscale turbulent field. The annual mean depth-integrated semidiurnal coherent energy flux for the regional model domain is shown in Fig. 3 suggesting that the glider crossed several times an area of pronounced westward internal tide energy propagation, characterized by narrow tidal beams. The temporal overlap of glider observations and numerical model output allows for a complementary assessment of both data sets.

### 2.3 Geostrophic surface currents

We used altimetry-derived global ocean gridded maps (1/4°) of geostrophic surface currents from absolute dynamic topography, generated and processed by the E.U. Copernicus Marine Environment Monitoring Service (CMEMS). We used the multimission Data Unification and Altimeter Combination System (DUACS) product in delayed time and daily resolution with all satellite missions available at a given time. We extracted geostrophic surface currents for the glider period August-October 2014, which served as input for the ray tracing in Sect. 3.5.

### 2.4 HRET

The internal tide induced SSH signature along the glider track is computed using empirical estimates from the High Resolution Empirical Tide product (HRET version 8.1, Zaron, 2019). This product uses essentially all-exact repeat altimeter mission data during a 25-year long time series of SSH observations (1992-2017, TOPEX/Jason, Geosat, ERS, Envisat). Based on a pointwise harmonic analysis along all available satellite track, and a plane-wave fit in overlapping patches, it provides the coherent amplitude and phase for the major semidiurnal (M2, S2) and diurnal (K1, O1) tides for modes 1-3. Here, we reconstruct

the coherent semidiurnal timeseries of SSH in along glider-track direction using the M2 and S2 harmonics for modes 1-2 to compare with the glider derived steric height (Sect. 2.4).

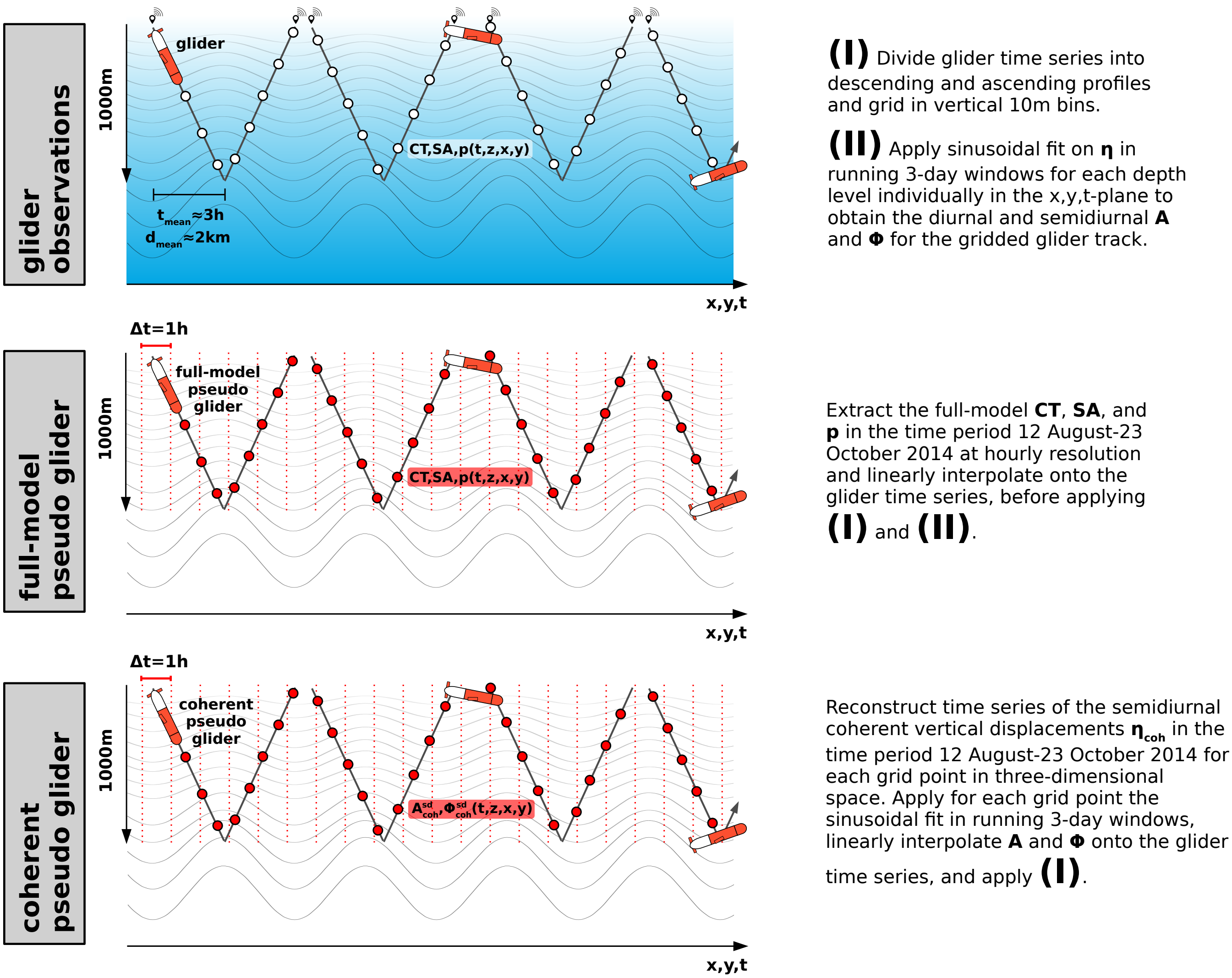

**Figure 4.** Schematic for the glider observations (top panel), the full-model pseudo glider (middle panel), and the coherent pseudo glider (bottom panel). Note that the full-model pseudo glider and the coherent pseudo glider sampling are identical to the one in the glider observations, i.e. the sampling considers the horizontal displacement during the glider profile as a function of time. This is achieved by linearly interpolating the model hourly output onto the glider time series.

## 3 Methods

### 3.1 Glider-derived internal tide amplitude and phase in 3-day running windows

The amplitude and phase of internal tides, derived from glider observations with irregular sampling in both space and time, are determined using a well-established methodology from Rainville et al. (2013). This methodology relies on the sinusoidal regression of vertical (isopycnal) displacements in 3-day running windows using a least squares fit. The choice of the 3-day window is ultimately linked to a compromise between 1) a minimum time series length that captures the diurnal period between 18 and 36 h and the semidiurnal period between 10.3 and 14.4 h, 2) an adequate number of tidal cycles within 3 days for the statistical analysis, and 3) a time period during which the internal tide amplitude and phase do not vary significantly during the glider's horizontal displacement. In our case, the corresponding travel distance over 3 days is approximately 50 km. The vertical displacement is computed as follows:

$$\eta = g\frac{\sigma_s - \overline{\sigma}}{\sigma_s \overline{N^2}}, \tag{1}$$

where g is the gravitational acceleration, $\sigma_s$ is the sample density, and $\overline{\sigma}$ and $\overline{N^2}$ are the mean density and mean squared buoyancy frequency relative to the 3-day running window, respectively. Further, a linear trend was subtracted which we attribute to sloping isopycnals of low-frequency motion. Following Rainville et al. (2013), we fit simultaneously the K1 ($\omega_{\mathrm{d}}$ = 2$\pi$/23.9345 h$^{-1}$) and M2 ($\omega_{\mathrm{sd}}$ = 2$\pi$/12.4206 h$^{-1}$) internal tide for each depth layer and each 3-day time window, i.e.

$$\eta(t,z) = A_{\mathrm{d}}(z)e^{(-iw_{\mathrm{d}}t+\phi_{\mathrm{d}}(z))} + A_{\mathrm{sd}}(z)e^{(-iw_{\mathrm{sd}}t+\phi_{\mathrm{sd}}(z))}, \tag{2}$$

where $A_{\mathrm{d}}$ and $A_{\mathrm{sd}}$ are the diurnal and semidiurnal amplitude, respectively. Equivalently, $\phi_{\mathrm{d}}$ and $\phi_{\mathrm{sd}}$ are the diurnal and semidiurnal phases. Here, the phase is relative to the Unix epoch (00:00:00 UTC on 1 January 1970). Note that even though we solve for the peak frequencies of the K1 and M2 tides, the fitted amplitude and phase are representative for the diurnal and semidiurnal frequency band since the 3-day window does not allow for a separation among the diurnal or semidiurnal tidal constituents. The sampling of the glider observations, the underlying methodology to extract the diurnal and semidiurnal tide, and the overall workflow is illustrated and summarized in the schematic in Fig. 4.

The sinusoidal fit for the diurnal and semidiurnal internal tide applied on the glider observations is explicitly shown for a 3-day window in Fig. 5. Explained variability $\gamma$ is here given by the covariance: cov($\eta_{\mathrm{d}}$,$\eta$)/var($\eta$) for the diurnal fit with the diurnal internal tide induced vertical displacement $\eta_{\mathrm{d}}$ and cov($\eta_{\mathrm{sd}}$,$\eta$)/var($\eta$) for the semidiurnal fit with the semidiurnal internal tide induced vertical displacement $\eta_{\mathrm{sd}}$. In this example, the measured signal is governed by a semidiurnal cycle, well captured by the sinusoidal fit and explaining 77 % of the total variance. The diurnal signal is rather weak in amplitude and barely contributes to the total variance (1 %). The residual signal accounts for 22 %. Overall, the methodology gives us confidence in our ability to accurately reconstruct the amplitude and phase of the internal tide south of New Caledonia.

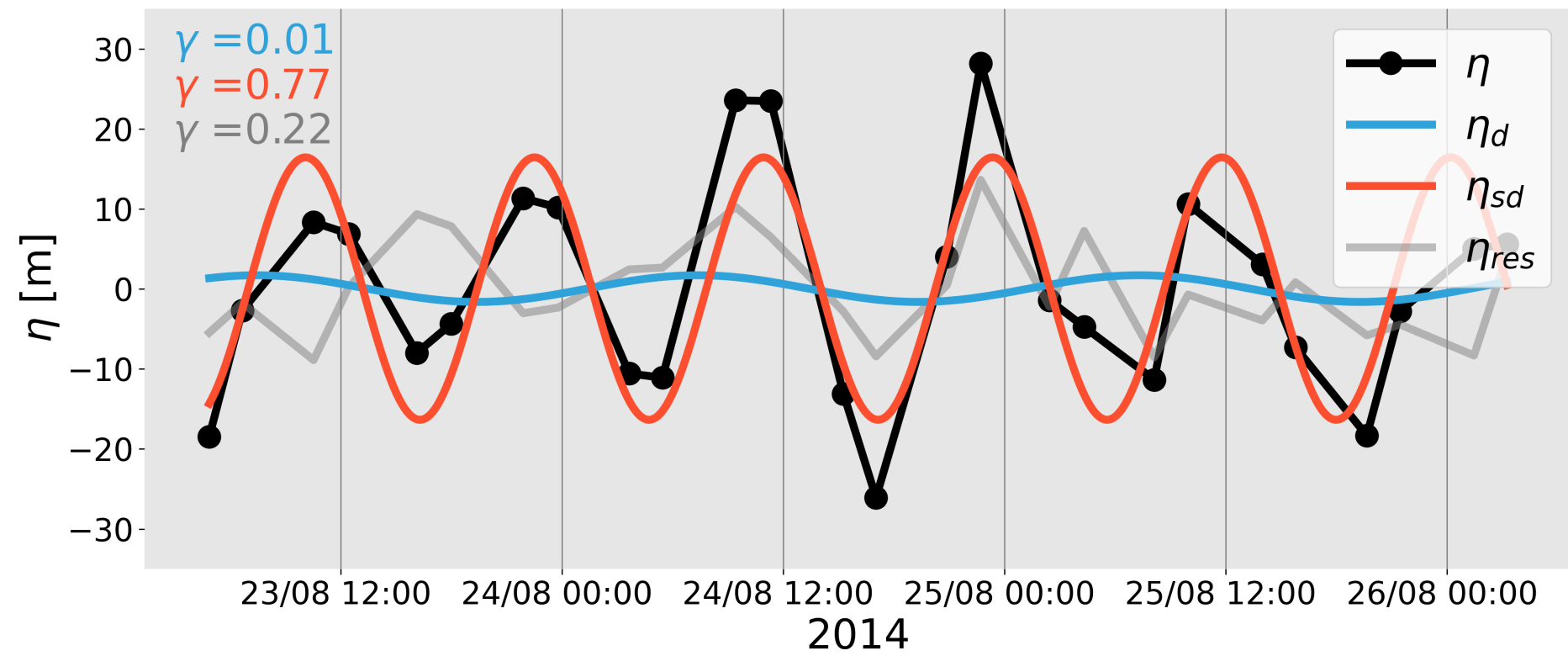

**Figure 5.** Glider observations derived vertical displacements (black, $\eta$) for an exemplary 3-day window at 300 m depth and the sinusoidal fit of the diurnal (blue, $\eta_\mathrm{d}$), semidiurnal (red, $\eta_\mathrm{sd}$) internal tide, and the residual signal (gray, $\eta_\mathrm{res}$). The respective explained variability $\gamma$ is also shown.

### 3.2 Full-model pseudo glider simulation

The temporal overlap of glider observations and numerical simulation output is a great opportunity to compare in-situ observations and the regional simulation. To do so, we simulate what we call a full-model pseudo glider by extracting the model's three-dimensional and hourly output for conservative temperature, salinity, and pressure (see middle panel in Fig. 4). The variables are then interpolated onto the glider track with the same irregular spatio-temporal sampling than the glider observations, followed by the division into descending and ascending profiles and the gridding in vertical 10 m bins. Moreover, diurnal and semidiurnal amplitude and phase are determined just as in the glider observations by applying a sinusoidal fit on vertical displacements in 3-day running windows. In this way, we create pseudo-like observations mimicking the glider mission with the full-model variability.

### 3.3 Coherent pseudo glider simulation

In this study we build upon a harmonic analysis performed on the full-model output for each grid point in the regional model domain as described in Bendinger et al. (2023). Briefly, the objective is to obtain a reference data set from a modeling perspective for the coherent internal tide amplitude and phase along the glider track. We refer to this as the coherent pseudo glider. The methodology is presented in the following and illustrated in the bottom panel in Fig. 4. The coherent internal tide induced amplitude and phase of vertical displacements are deduced from the three-dimensional baroclinic vertical velocity harmonic as follows:

$$\eta_\mathrm{coh}(z) = \frac{w_A(z)T}{2\pi}, \quad (3)$$

where $w_A$ is the harmonically fitted amplitude of vertical velocity for each grid point in the domain and $T$ is the respective tidal period. Note that here we only consider the semidiurnal coherent internal tide since it is largely dominant over the diurnal tide as shown further below. The semidiurnal coherent vertical displacements are computed by reconstructing a time series for each semidiurnal tidal constituent (i.e. M2, S2, N2) using $w_A$ and the harmonically fitted phase $w_\Phi$, before summing over the three time series to obtain the semidiurnal time series $\eta^{\mathrm{sd}}_{\mathrm{coh}}$ for each grid point in three-dimensional space. Once reconstructed, we apply the sinusoidal fit in 3-day running windows just as we did for the glider observations, but for each grid point. We obtain the semidiurnal coherent amplitude $A^{\mathrm{sd}}_{\mathrm{coh}}$ and phase $\Phi^{\mathrm{sd}}_{\mathrm{coh}}$ in three-dimensional space at hourly resolution. Finally, we interpolate $A^{\mathrm{sd}}_{\mathrm{coh}}$ and $\Phi^{\mathrm{sd}}_{\mathrm{coh}}$, onto the glider track with the same irregular spatio-temporal sampling than the glider observations, and as above followed by the division into descending and ascending profiles and the gridding in vertical 10 m bins. We computed the semidiurnal coherent amplitude and phase for both the total baroclinic signal (modes 1-9) and mode 1.

### 3.4 Climatological vertical modes

We computed climatological vertical mode profiles for vertical velocity and displacement along the glider track to infer the modal structure of the glider and the pseudo glider vertical displacements (see Sect. 3.6.1). Climatological modes are computed by solving the Sturm-Liouville eigenvalue problem (Gill, 1982):

$$\frac{\partial^2 \Phi_n}{\partial z^2} + \frac{N^2}{c_n^2}\Phi_n = 0, \tag{4}$$

where $\Phi$ is the eigenfunction describing the vertical structure for vertical velocity or displacement subject to the boundary conditions $\Phi(0) = \Phi(-H) = 0$, $n$ is the mode number, and $c_n$ is the separation constant. We solve the eigenvalue problem for climatological profiles of stratification $N$ inferred from climatological profiles of conservative temperature and absolute salinity taken from the CSIRO Atlas of Regional Seas (CARS) for each glider profile location (Ridgway et al., 2002). For this study's purpose, the climatological modes were averaged along the glider track and cut to a representative depth of 3000 m, which is being considered as the full-depth range below.

### 3.5 Ray tracing

A ray tracing method following Rainville and Pinkel (2006) is used to infer the departure from tidal coherence in the glider observations or full-model pseudo glider, associated with the refraction of the tidal beam due to mesoscale background currents, i.e. mesoscale eddies. Specifically, the horizontal propagation of internal gravity modes is investigated considering spatially varying topography, climatological stratification, planetary vorticity, and depth-independent currents. Following this approach, we assume that departure from tidal coherence is primarily due to varying background currents. The choice of depth-independent currents relies on the general assumption that mesoscale eddies are well represented by a barotropic and a mode-1 baroclinic structure with limited vertical shear (Smith and Vallis, 2001). Also, the assumption of considering only depth-independent is validated a posteriori, given the relevance of the qualitative picture of ray trajectories that are obtained.

Bathymetry is taken from ETOPO2v2 (Smith and Sandwell, 1997). Internal gravity wave speeds are predefined and solved by the Sturm-Liouville problem for stratification from the World Ocean Atlas (Locarnini et al., 2018; Zweng et al., 2019). We model semidiurnal ray paths for modes 1-2, initialized at the internal tide generation hot spot south of New Caledonia near Pines Ridge (167.65° E, 23.35° S) and for a given propagation angle (southwestward; 210°). In an iterative procedure, the ray tracing considers for each step size (1 km) bathymetry, climatological buoyancy and planetary vorticity effects, and the background currents. Through the dispersion relation from the Helmholtz equation for internal wave modes assuming a local wave expression, the ray's group and phase velocity are obtained, which are then used to update the ray's position and direction (angle of propagation). To mimick the effects of background currents on the ray's path, a no-currents scenario is also given. The ray tracing is applied on two different velocity products: (1) the depth-averaged currents as derived from the daily-mean three-dimensional velocity field from the regional model output (CALEDO60) and (2) the geostrophic surface currents as derived from CMEMS. For the latter, the depth-independent currents are approximated by multiplying the SSH derived geostrophic surface currents by a factor of 0.5, following Rainville and Pinkel (2006). Note that the ray tracing results of (1) and (2) are rather qualitative. The two data products used in the ray tracing contain different dynamics. Specifically, the gridded two-dimensional fields from CMEMS do not resolve the same spatial and temporal scales as CALEDO60. Thus, the direct comparison of the ray tracing needs to be taken with caution.

### 3.6 Internal tide induced steric height

To our knowledge, glider observations have never been used the SSH signature of fine-scale dynamical structures such as internal tides. In the following, we introduce the methodology to derive the SSH signature, i.e. steric height, of internal tides using glider data limited to the upper 1000 m. Further, the available data in the upper 1000 m is exploited to account for the internal tide steric height signal at depths beyond 1000 m and, thus, for the full-depth range.

#### 3.6.1 Glider and pseudo glider steric height

Following Zhao et al. (2010), we deduce the steric height $h$ of internal tides from surface pressure $p_{\mathrm{surf}}$, i.e. the vertical integral of vertical displacements $\eta$:

$$h(t) = \frac{1}{\rho_0 g} \underbrace{\rho_0 \int_{-H}^{0} N^2(z,t)\eta(z,t)dz}_{p_{\mathrm{surf}}(t)}, \tag{5}$$

where $H$ is the ocean depth. Since the glider observations are limited to the upper 1000 m depth, Equation 5 becomes:

$$h_{\mathrm{obs}}^{1000\mathrm{m}}(t) = \frac{1}{\rho_0 g} \underbrace{\rho_0 \int_{1000m}^{0} N^2(z,t)\eta_{\mathrm{obs}}^{1000\mathrm{m}}(z,t)dz}_{p_{\mathrm{surf}}^{1000\mathrm{m}}(t)}, \tag{6}$$

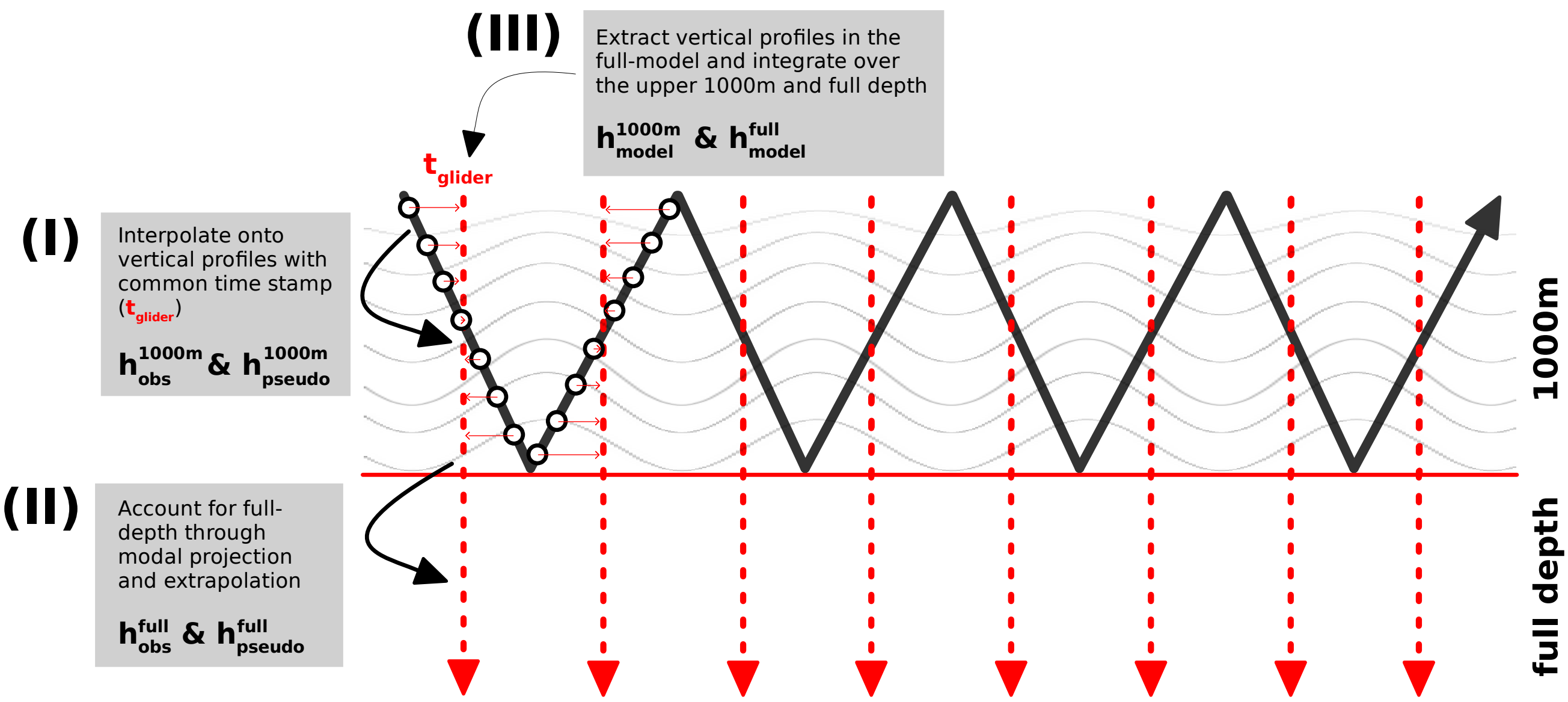

**Figure 6.** Schematic illustrating (I) the linear interpolation of vertical displacements for a given profile onto a common time stamp $t_{\text{glider}}$ prior to the vertical integration to obtain $h^{1000\text{m}}_{\text{obs}}$ and $h^{1000\text{m}}_{\text{pseudo}}$, (II) the modal projection on climatological modes and extrapolation to the full-depth range to obtain $h^{\text{full}}_{\text{obs}}$ and $h^{\text{full}}_{\text{pseudo}}$, and (III) the vertical extraction in the full-model at $t_{\text{glider}}$ to obtain $h^{1000\text{m}}_{\text{model}}$ and $h^{\text{full}}_{\text{model}}$.

where $h^{1000\text{m}}_{\text{obs}}$ is the glider steric height and $\eta^{1000\text{m}}_{\text{obs}}$ are the glider vertical displacements. Since glider measurements for a given profile are not instantaneous and are associated with different time stamps, we interpolated the vertical displacements onto a common time stamp ($t_{\text{glider}}$) that represents the glider's time stamp at mid-depths (see the schematic in Fig. 6). This is achieved by interpolating the reconstructed time series, utilizing amplitude and phase data from the sinusoidal fit, onto the respective $t_{\text{glider}}$ of the profile.

Additionally, we explore steric height inferred by a modal projection of glider vertical displacements on a set of climatological modes followed by an extrapolation to the full-depth range. The objective is to understand to what extent Equation 6 and, thus, the vertical integral limited to glider measurements in the upper 1000 m is a sufficient approximation to account for the surface signature of internal tides. Further, we investigate whether we could infer the surface signature with a better accuracy assuming that internal tides are well represented by the modal structure of modes 1-2.

Using a least-squares fitting method, we project the glider vertical displacements onto a set of climatological modes, as obtained from Sect. 3.4 and limited to the upper 1000 m:

$$\eta^{1000\text{m}}_{\text{obs}}(z,t) = \sum_{n=1}^{2} \widetilde{\eta}^{1000\text{m}}_{n,obs}(t)\Phi^{1000\text{m}}_{n}(z), \tag{7}$$

For each time step, the least-squares solution is then used to extrapolate to the full-depth range using the regression coefficient $\widetilde{\eta}_{n,obs}^{1000\mathrm{m}}$ and the full-depth climatological mode $\Phi_n^{\mathrm{full}}$ to obtain the full-depth glider vertical displacements $\eta_{\mathrm{obs}}^{\mathrm{full}}$. In this study the glider vertical displacements are projected using a maximum of two modes (two-mode approximation). The projection on the first mode only is referred to as first-mode approximation. The associated steric height for the full-depth range was computed equivalent to above:

$$h_{\mathrm{obs}}^{\mathrm{full}}(t) = \frac{1}{\rho_0 g} \underbrace{\rho_0 \int_{-H}^{0} N^2(z)\eta_{\mathrm{obs}}^{\mathrm{full}}(z,t)dz}_{p_{\mathrm{surf}}^{\mathrm{full}}(t)}. \tag{8}$$

where $h_{\mathrm{obs}}^{\mathrm{full}}$ is referred to as the full-depth glider steric height. Similarly, we compute pseudo glider steric height $h_{\mathrm{pseudo}}^{1000\mathrm{m}}$ and the full-depth pseudo glider steric height $h_{\mathrm{pseudo}}^{\mathrm{full}}$ as deduced from $\eta_{\mathrm{pseudo}}^{1000\mathrm{m}}$ and $\eta_{\mathrm{pseudo}}^{\mathrm{full}}$, respectively.

#### 3.6.2 Full-model steric height

To have a ground truth for the internal tide surface signature for the full-depth range, we computed steric height from the regional model output as follows:

$$h_{\mathrm{model}}^{\mathrm{full}} = \int_{-H}^{\eta_0} \delta(z)\rho_0 dz, \tag{9}$$

where $\rho_0$=1035 kg m$^{-3}$ is the reference density, $\eta_0$ is the free surface displacement and $\delta$ is the specific volume anomaly. The latter is computed as:

$$\delta(z) = \frac{1}{\rho(\mathrm{CT},\mathrm{SA},z)} - \frac{1}{\rho(\mathrm{CT}_{\mathrm{ref}},\mathrm{SA}_{\mathrm{ref}},z)} \tag{10}$$

with $\mathrm{CT}_{\mathrm{ref}}$ and $\mathrm{SA}_{\mathrm{ref}}$ the reference conservative temperature (0° $C$) and the reference absolute salinity (Standard Ocean Reference Salinity, 35.16504 g/kg). Steric anomaly was computed for each grid point from vertical profiles of conservative temperature and absolute salinity. The steric SSH is then obtained through vertical integration before being bandpassed in the semidiurnal frequency band and interpolated onto the time stamps representative for the glider profiles along the glider track ($t_{\mathrm{glider}}$ in Fig. 6). The full-depth steric height $h_{\mathrm{model}}^{\mathrm{full}}$ is compared with the full-depth pseudo glider steric height $h_{\mathrm{pseudo}}^{\mathrm{full}}$. In addition, we also computed the steric height contribution in the upper 1000 m, i.e.

$$h_{\mathrm{model}}^{1000\mathrm{m}} = \int_{1000m}^{\eta_0} \delta(z)\rho_0 dz, \tag{11}$$

which serves for the comparison with the pseudo glider steric height $h_{\mathrm{pseudo}}^{1000\mathrm{m}}$.

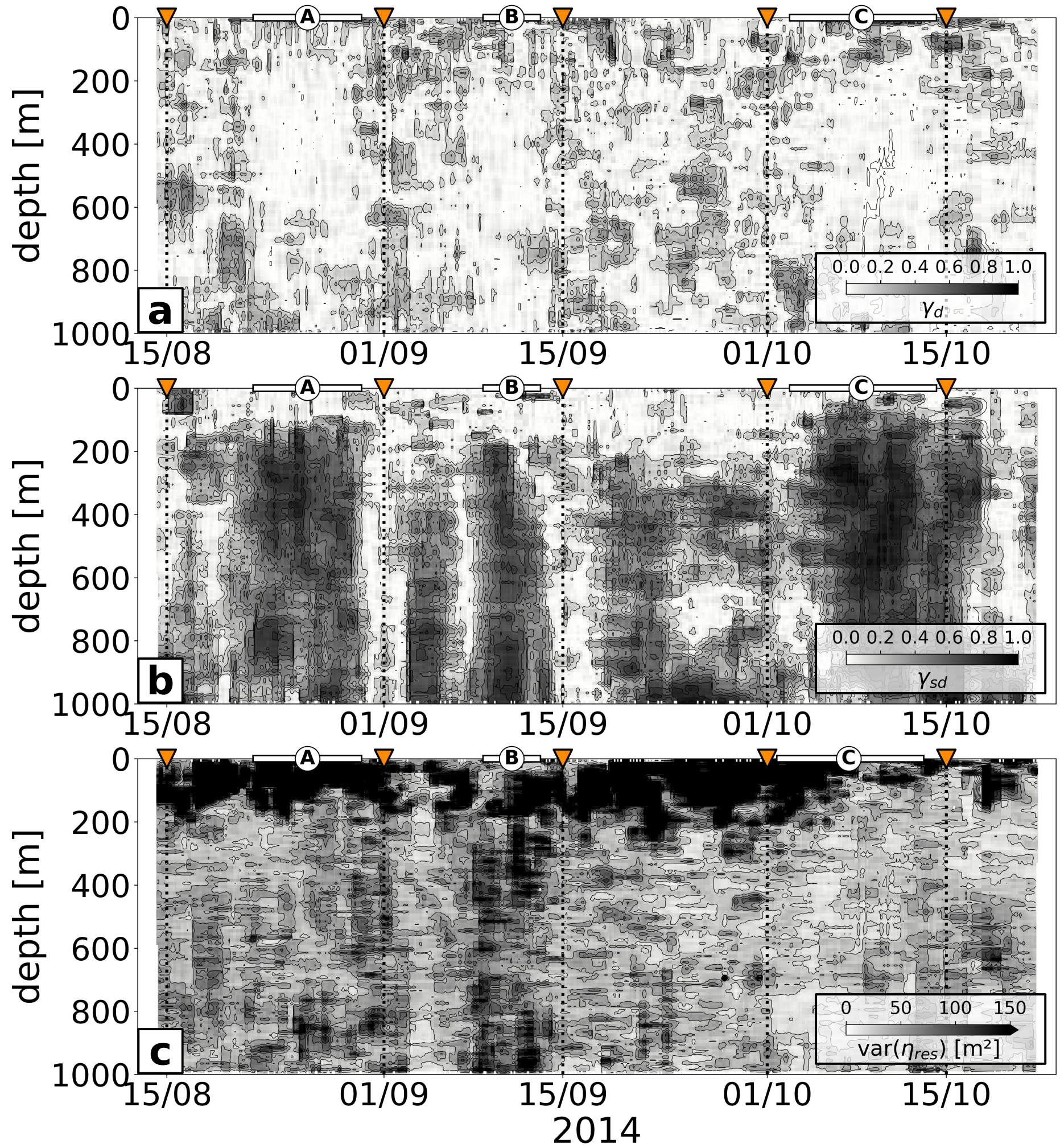

**Figure 7.** Explained variance by the (a) diurnal ($\gamma_\mathrm{d} = \mathrm{cov}(\eta_\mathrm{d}, \eta)/\mathrm{var}(\eta)$) and (b) semidiurnal ($\gamma_{sd} = \mathrm{cov}(\eta_\mathrm{sd}, \eta)/\mathrm{var}(\eta)$) internal tide as given by the sinusoidal fit applied on the glider observations. (c) Variance of the residual signal $\mathrm{var}(\eta_\mathrm{res})$.

## 4 Results

In the following, we first explore the glider observations, the underlying dominance and spatio-temporal variability of the semidiurnal internal tide before comparing it with the model. The pseudo glider simulation is then used to link the ocean interior with the ocean surface by investigating the extent to which the upper 1000 m vertical displacements are sufficient to determine the steric height signal of internal tides.

### 4.1 Semidiurnal internal tide dominance

The simultaneous fit of vertical displacements $\eta$ for the diurnal and semidiurnal tide for the upper 1000 m along the whole glider section, expressed by the explained variance of the total signal, is presented in Fig. 7. The field is overwhelmingly dominated by the semidiurnal tide, along the entirety of the section and throughout the water column from 100-200 m below the surface down to 1000 m depth. Locally, the semidiurnal fit explains up to 80 % of the total variance (Fig. 7b). The diurnal tide is of rather patchy nature and explains less than 10 % of the whole signal most of the time (Fig. 7a). The residual signal stands out as it features high variance near the surface in the upper 100-200 m while diurnal and semidiurnal signals are weak. This corresponds with expectations, i.e. vertical displacements of low-mode baroclinic tides are zero at the surface and weak in the upper layers. Further, the mixed layer near the surface represents a complex flow regime with a broad variety of dynamics, i.e. internal tide induced vertical displacements are no longer the dominant signal. Considering that the fit is performed over a period of 3 days and a horizontal distance of 50 km, the residual may be linked to surface-intensified submesoscale dynamics. In the following, we will only focus on the semidiurnal internal tide since it represents the dominant signal in the region. This is also in agreement with Bendinger et al. (2023), which attributed 96 % of the barotropic-to baroclinic energy conversion in their regional model domain to the semidiurnal tide (M2, S2, N2).

### 4.2 Spatio-temporal variability of semidiurnal internal tide: in-situ observations vs. numerical model

The spatio-temporal variability of the semidiurnal internal tide is investigated in the following (Fig. 8). The glider observations reveal strong spatio-temporal variability during the >2 months glider survey as shown by the semidiurnal amplitude (Fig. 8a) and phase (Fig. 8e). Along the glider track, there are distinct patterns of enhanced semidiurnal tide amplitude (>20 m) corresponding to large fractions of explained variability (see Fig. 7b).

Based on the modeling results, these patterns are localized to the most distinct tidal beams (labeled as A, B, and C). At these locations, semidiurnal tidal energy propagates westward, which can be traced back to the formation and/or superposition of tidal beams to the south/southeast of New Caledonia (Fig. 3). The pronounced southwestward propagating tidal beam is crossed twice: on the way south in late August (A), and when heading back north in early to mid-October (C) towards the coast of New Caledonia. A third distinct tidal beam worth mentioning is encountered when the glider changes its heading direction from southward to northward (B). Another double crossing of tidal beams (but with weaker amplitudes) is observed in early September and from mid-to-end of September, also visible in the depth-integrated semidiurnal energy flux (Fig. 3).

Glider observations and the full-model pseudo glider show an overall similarity in the spatio-temporal variability. Specifically, this applies to the location, the magnitude, and the vertical structure/extent of the tidal beams (Fig. 8a,e for the glider observations and Fig. 8b,f for the full-model pseudo glider).

Differences between the observations and the full-model pseudo glider are most evident mid-August and early October. During these periods, we find strong tidal signatures in the full-model pseudo glider expressed by elevated amplitudes of isopycnal displacements. This is also apparent from the fitted phase. Given that both glider observations and the full-model pseudo glider feature identical sampling, variations linked to the spring-neap tide cycle are not valid hypotheses. Discrepancies may arise due to inaccuracies in simulating the precise beam locations or in deficiencies in representing the model's vertical mode structure. Another contributing factor is tidal incoherence, which can arise from eddy-internal tide interactions, i.e. temporally varying

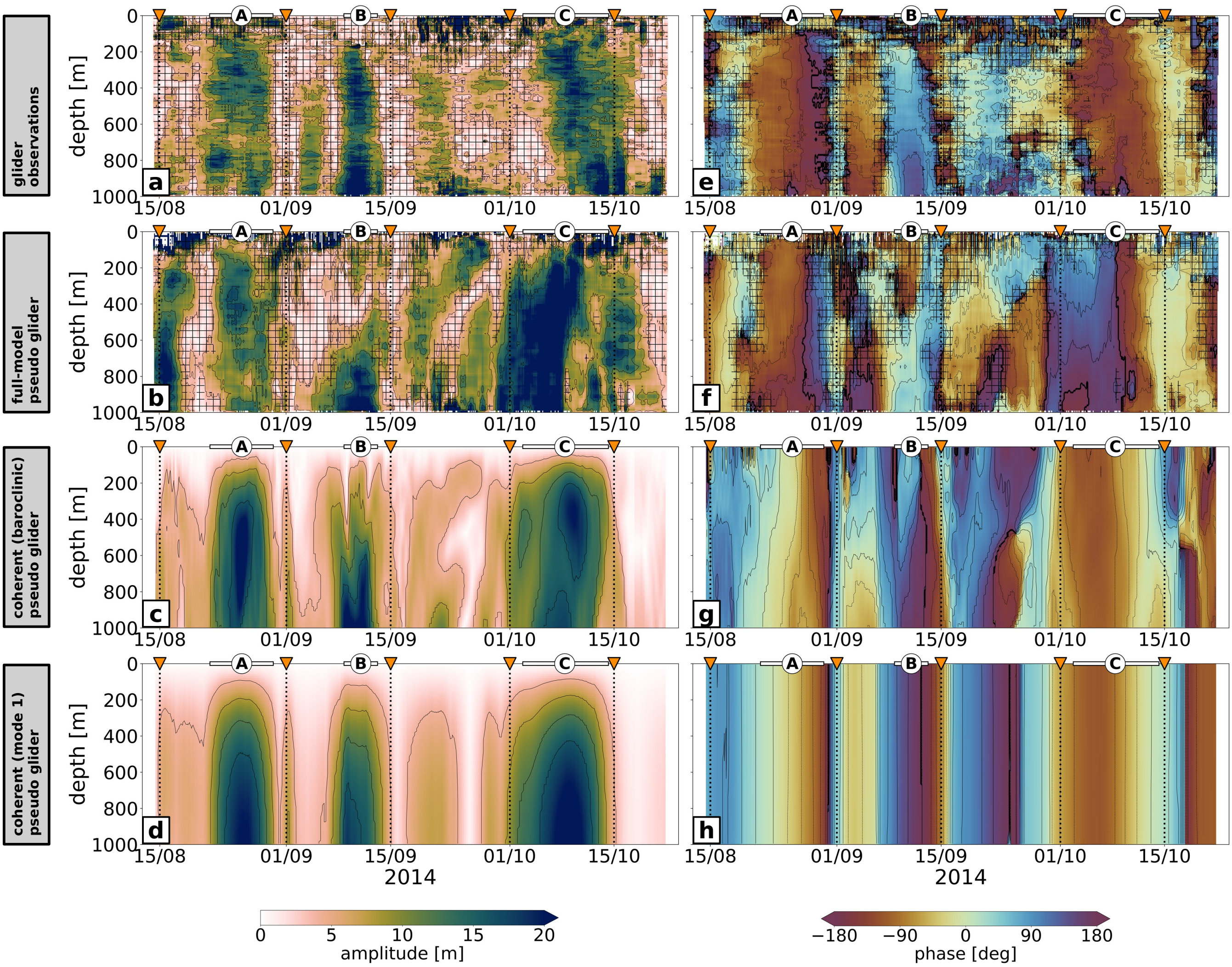

**Figure 8.** Semidiurnal amplitude for (a) glider observations, (b) the full-model pseudo glider, and (c) the coherent pseudo glider. (d) Same than (c) but for vertical mode 1. (e-f) Same than (a-d) but for the semidiurnal phase. Hatches in (a-b) and (e-f) represent areas where the residual signal explains more than twice the variance of the semidiurnal fit.

background currents. Mesoscale and submesoscale features are by nature stochastic. Particularly, mesoscale and submesoscale eddies are not expected to be found at similar location and time in reality and the numerical simulation. The semidiurnal internal tide as derived from glider observations and the full-model pseudo glider appears to a large extent of coherent nature when taking the coherent pseudo glider as a reference data set (Fig. 8c,g). The tidal signatures mid-August and early October in the full-model pseudo glider pose a clear exception. In the subsequent section, we explore whether discrepancies in the mesoscale background field offer insights into differences observed in the sampled semidiurnal internal tide field.

### 4.3 Tidal incoherence inferred from pseudo glider simulation

To infer the impact of mesoscale eddies on the tidal beam's refraction and corresponding incoherence, we apply a simplified ray tracing following Rainville and Pinkel (2006), see Sect. 3.5. We initiate a semidiurnal ray just west of the Pines Ridge, which is known as a hot spot of internal tide generation (Bendinger et al., 2023). The theoretical ray paths for modes 1-2 are shown for two different snapshots on 13 August 2014 (Fig. 9) and 3 October 2014 (Fig. 10). The background velocity field is clearly different between glider observations and the full-model pseudo glider for both snapshots. The no-currents scenario is by definition the same since it relies on the same climatological stratification, bathymetry, and planetary vorticity. The semidiurnal ray

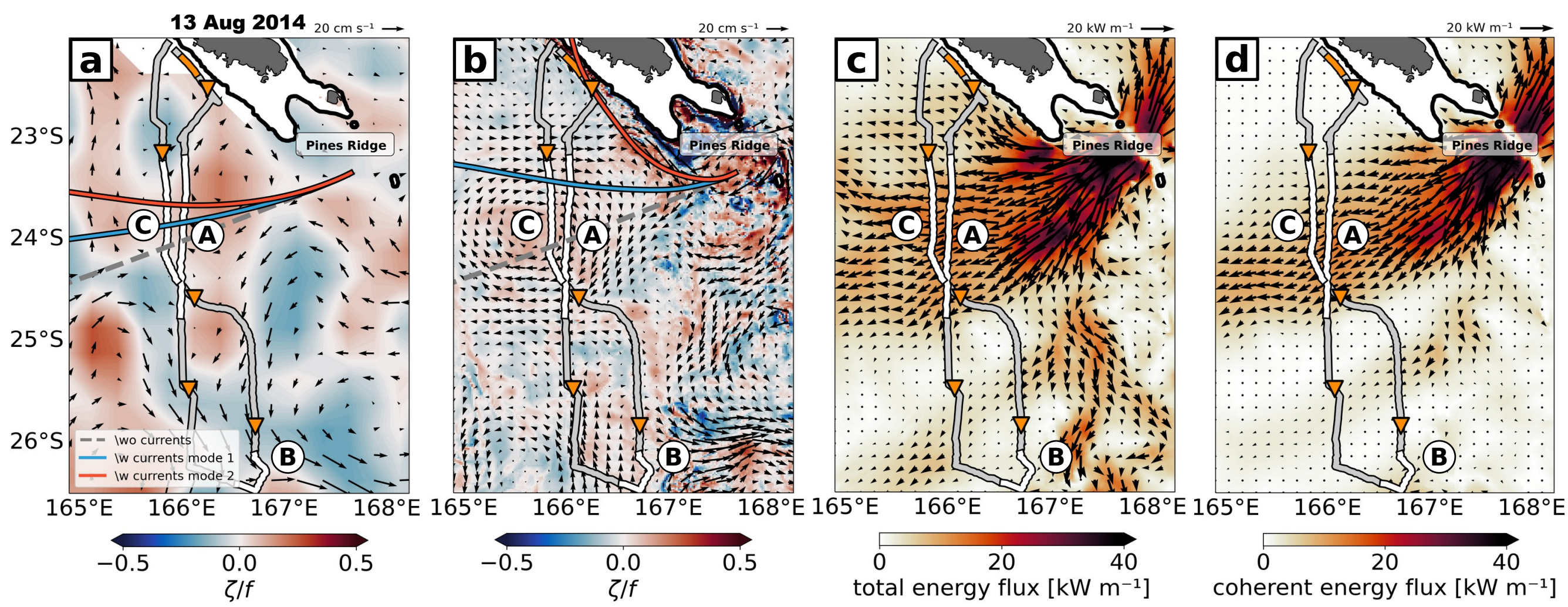

**Figure 9.** Zoom into the glider study site showing (a) the geostrophic surface currents (vectors) as obtained from satellite altimetry (CMEMS) with the underlying relative vorticity normalized by $f$ (shading) for a daily-mean snapshot on 13 August 2014. (b) Same as (a) but showing the depth-averaged currents as obtained from CALEDO60. Also shown is the modeled (c) total semidiurnal energy flux, and (d) coherent semidiurnal energy flux for a daily-mean snapshot on 13 August 2014. The ray-tracing results in (a) and (b) show the theoretical ray path of a semidiurnal tidal beam for mode 1 (blue), mode 2 (red) that initiate from the generation site south of New Caledonia close to the Pines Ridge. The no-current scenario is also given (dashed gray). The orange triangles are as in Fig. 3. The glider position and the distance covered on 13 August 2014 are shown by highlighted orange bar along the glider track. The thick black line is the 100 m depth contour representative for the New Caledonian lagoon.

propagates southwestward corresponding to the well-confined propagation direction of the coherent tidal beam. Including the background currents introduces differing ray paths between glider observations and the full-model pseudo glider, as explored next.

On 13 August 2014, the mesoscale eddy field as given by satellite altimetry derived geostrophic surface currents has only little impact on the semidiurnal ray path, though it is slightly refracted northward. Mode 2 is more affected than mode 1 (Fig. 9a). This contrasts with the numerical simulation, which is characterized by a mesoscale cyclone close to the New Caledonia coast and the Pines Ridge, and which quickly refracts the semidiurnal ray northward (Fig. 9b). This aligns with the modeled semidiurnal energy flux along the western New Caledonia coast (Fig. 9c) and corresponds with the full-model pseudo glider sampling at the start of the section in mid-August (see Fig. 8b,f).

On 3 October 2014, the ray tracing provides a similar picture (Fig. 10). Even though a mesoscale anticyclone governs the background velocity field in the satellite altimetry observations, the tidal beam orientation is barely affected (Fig. 10a). Further, it is refracted equatorward, away from the glider sampling and therefore not captured by the glider observations early October (see Fig. 8a,e). This is opposed to the numerical simulation in which a predominant anticyclone is located just off the New Caledonia coast and the Pines Ridge (Fig. 10b). In propagation direction, the semidiurnal rays are increasingly refracted southward (poleward) with increasing distance to the initialization region as they pass through the mesoscale eddy. Again, mode 2 is more affected than mode 1. Particularly, the theoretical beam is refracted toward the full-model pseudo glider sampling

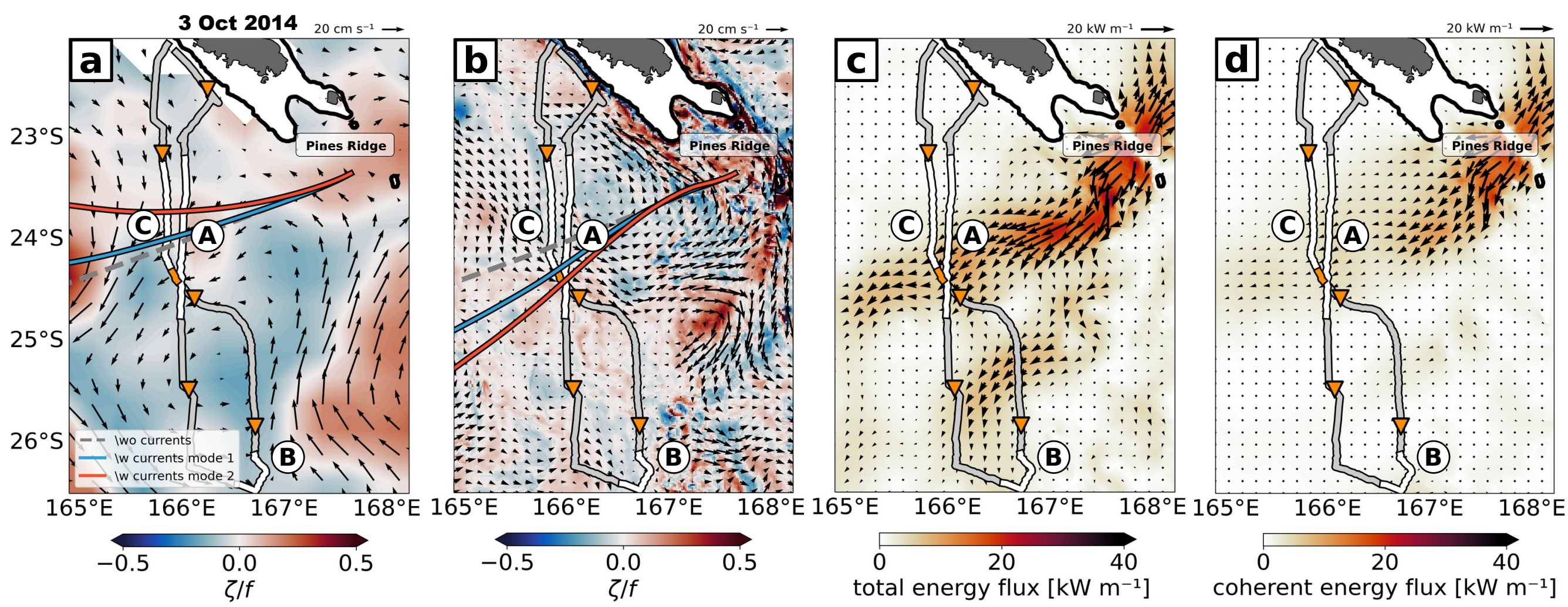

**Figure 10.** Same as Fig. 9, but for a daily-mean snapshot on 3 October 2014. Note that the semidiurnal energy flux underlies spring-neap tide variability, which explains the differences in the modeled coherent energy flux for different snapshots. The difference between the total and coherent energy flux can be associated with tidal incoherence.

**Table 1.** Explained variability of $h^{\mathrm{1000m}}_{\mathrm{model}}$, $h^{\mathrm{1000m}}_{\mathrm{pseudo}}$, $h^{\mathrm{full}}_{\mathrm{model}}$ using a first-mode approximation ($\Sigma^{1}_{n=1}$) and a two-mode approximation ($\Sigma^{2}_{n=1}$), referenced to $h^{\mathrm{full}}_{\mathrm{model}}$.

| | $h^{\mathrm{1000m}}_{\mathrm{model}}$ | $h^{\mathrm{1000m}}_{\mathrm{pseudo}}$ | $h^{\mathrm{full}}_{\mathrm{pseudo}}\Sigma^{1}_{n=1}$ | $h^{\mathrm{full}}_{\mathrm{pseudo}}\Sigma^{2}_{n=1}$ |
|---|---|---|---|---|
| $\gamma$ (cov(...,$h^{\mathrm{full}}_{\mathrm{model}}$)/var($h^{\mathrm{full}}_{\mathrm{model}}$) | 0.91 | 0.78 | 0.85 | 0.93 |

location, providing a possible explanation for the premature detection of the tidal beam early October (see Fig. 8b,f), which in the no-currents scenario is predicted at later stage $O(1\ \mathrm{d})$ along the glider track.

From an in-situ perspective, the glider observations have provided first insight into the spatio-temporal variability of the internal tide south of New Caledonia. Largely dominated by the semidiurnal tide, and especially mode 1, the glider observations also highlight the realism of internal tides in the regional numerical simulation in the upper 1000 m. We conclude that major discrepancies between glider observations and the full-depth pseudo glider are not associated with deficiencies in the glider sampling or the model's realism of simulating internal tides, but rather tidal incoherence. Here, we showed that tidal incoherence is largely depending on the background eddy field. In the following, we attempt to derive how the vertical structure of the semidiurnal internal tide in the upper 1000 m expresses in the steric SSH signature.

### 4.4 To what extent can we use gliders to account for internal tide induced steric height?

The ability of the glider to retrieve the steric SSH signature of the semidiurnal tide is investigated using the regional numerical simulation only. We first use the full-model pseudo glider simulation to evaluate the limitations due to the depth extent of the glider measurements. We compare (1) steric height inferred from the semidiurnal vertical displacements within the upper 1000 m only with, ($h^{\mathrm{1000m}}_{\mathrm{pseudo}}$) with (2) steric heigth inferred from the semidiurnal vertical displacements in the upper 1000 m extrapolated to full-depth using projections on climatological modes 1-2 ($h^{\mathrm{full}}_{\mathrm{pseudo}}$). To evaluate the errors in steric SSH reconstruction arising from the irregular glider sampling, we finally use the full-model steric SSH, which is free of any glider-like sampling. We remind the reader, that the full-model steric SSH was initially computed from vertical profiles of specific volume anomaly, integrated over the water column, bandpassed in the semidiurnal frequency band, and interpolated onto the glider's profile location and time stamp at mid-depth. It can be regarded as ground truth for steric SSH within the upper 1000 m ($h^{\mathrm{1000m}}_{\mathrm{model}}$), and the full-depth range ($h^{\mathrm{full}}_{\mathrm{model}}$). They are respectively confronted with $h^{\mathrm{1000m}}_{\mathrm{pseudo}}$ and $h^{\mathrm{full}}_{\mathrm{pseudo}}$ to assess the glider's ability to deduce steric SSH (see Sect. 3.6).

We first analyze $h^{\mathrm{full}}_{\mathrm{model}}$ along the glider track (Fig. 11a). The semidiurnal internal tide expresses in steric height with magnitudes of up to 5 cm. It is primarily associated with the locations that have been identified as segments of elevated tidal activity (A, B, C in Fig. 3 and Fig. 8). The overwhelming majority of the semidiurnal steric SSH, i.e. 91 % ($h^{\mathrm{1000m}}_{\mathrm{model}}$), is attributed to the contribution of internal tide vertical displacements in the upper 1000 m (Fig. 11a and Table 1).

The pseudo glider steric height ($h^{\mathrm{1000m}}_{\mathrm{pseudo}}$) follows the overall pattern of $h^{\mathrm{1000m}}_{\mathrm{model}}$ (Fig. 11a). However, it only accounts for 78 % of explained variance when referenced to $h^{\mathrm{full}}_{\mathrm{model}}$ (Table 1). This is, by construction, only due to the irregular glider sampling.

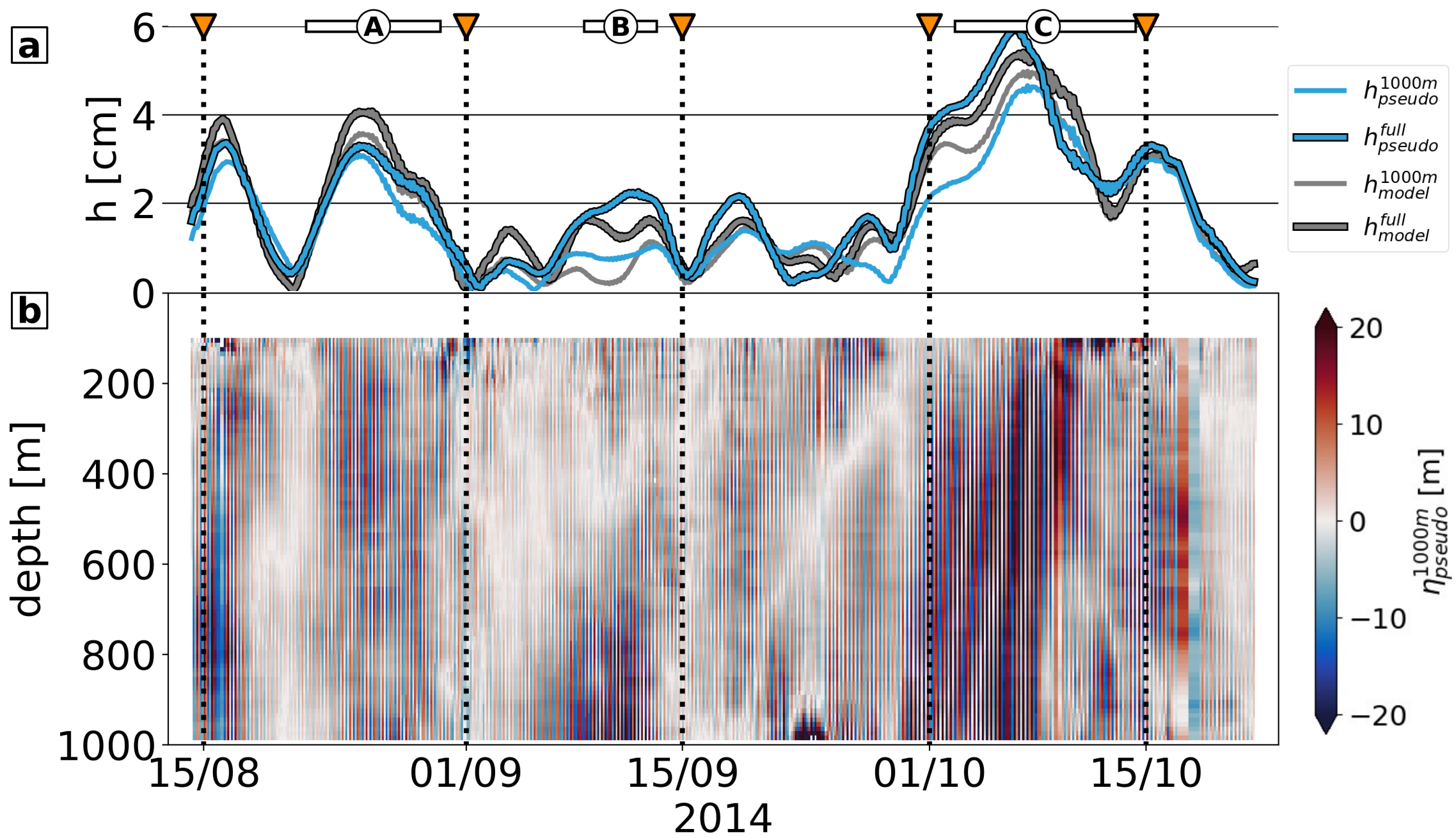

**Figure 11.** (a) Semidiurnal amplitude for the pseudo glider steric height $h^{\mathrm{1000m}}_{\mathrm{pseudo}}$ (blue), the full-depth pseudo glider steric height $h^{\mathrm{full}}_{\mathrm{pseudo}}$ (blue with black outlines), and the full-model steric height $h^{\mathrm{1000m}}_{\mathrm{model}}$ for the upper 1000 m (gray) and $h^{\mathrm{full}}_{\mathrm{model}}$ for the full-depth range (gray with black outlines. Here, we use a two-mode approximation to calculate $h^{\mathrm{full}}_{\mathrm{pseudo}}$. (b) Pseudo glider vertical displacements $\eta^{\mathrm{1000m}}_{pseudo}$ which are ultimately used to compute $h^{\mathrm{1000m}}_{\mathrm{pseudo}}$ and $h^{\mathrm{full}}_{\mathrm{pseudo}}$.

Projecting the pseudo-glider vertical displacements onto climatological modes and extrapolating to the full-depth range can partly account for missing variance due to the glider sampling limited to the upper 1000 m. Using a two-mode approximation, we are able to increase the explained variance from 78 % ($h^{\mathrm{1000m}}_{\mathrm{pseudo}}$) to 93 % ($h^{\mathrm{full}}_{\mathrm{pseudo}}\Sigma^{2}_{n=1}$, Table 1). From a qualitative perspective, this is also apparent in Fig. 11a, where $h^{\mathrm{full}}_{\mathrm{pseudo}}$ provides an overall improvement and compares better with $h^{\mathrm{full}}_{\mathrm{model}}$. Projecting on the two lowest modes gives the best result. Though, projecting on the first mode only increases the variance from 78 % to 85 % ($h^{\mathrm{full}}_{\mathrm{pseudo}}\Sigma^{1}_{n=1}$,Table 1).

### 4.5 How do the glider measurements compare with satellite altimetry?

Using a two-mode approximation, the methodology described in Sect. 3.6.1 and applied to the full-model pseudo glider in the previous section is now applied to the glider observations. The full-depth glider steric height $h^{\mathrm{full}}_{\mathrm{obs}}$ is compared to the empirical SSH estimates from HRET (see Sect. 2.4) to evaluate the spatio-temporal variability of the semidiurnal internal tide from a satellite altimetry point of view (Fig. 12). Note that in contrast to the glider observations, HRET represents exclusively the coherent tide. Within this context, we utilize HRET to reconstruct the semidiurnal signals composed of the M2 and S2 tides,

for modes 1-2, prior to interpolating the resulting time series to align with the time stamps along the glider's track.

The overall spatio-temporal variability of the full-depth glider steric height corresponds well with HRET (Fig. 12a and Fig. 12c). Particularly, the spatio-temporal signals in the segments denoted as A and C are well captured, even though reduced in amplitude (maximum 2 cm in HRET compared to maximum 4 cm in the glider observations). Good agreement is also seen for the signature mid-to-end September. Contrarily, the distinct signal, denoted as B, which exhibits a surface signature of almost

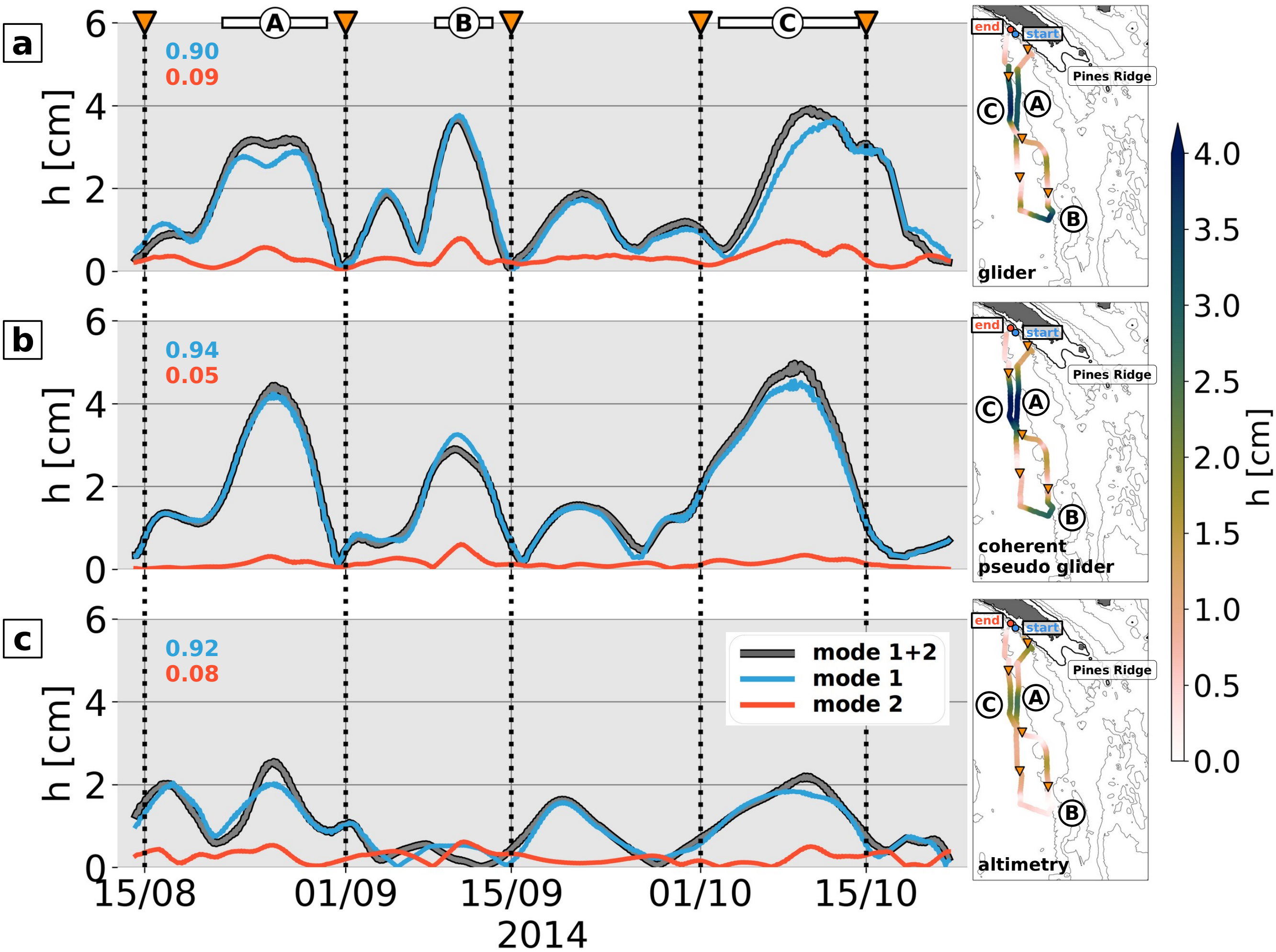

**Figure 12.** Semidiurnal amplitude in along glider-track direction of (a) the full-depth glider steric height using a two-mode approximation ($h_{\mathrm{obs}}^{\mathrm{full}}\Sigma_{n=1}^{2}$), (b) the full-depth coherent pseudo glider steric height $h_{\mathrm{coh}}^{\mathrm{full}}$, and (c) HRET SSH for the baroclinic tide (gray with black outlines), mode 1 (blue), and mode 2 (red). The modal contribution of modes 1-2 are given in the upper left corners. The right panels of (a)-(c) show the baroclinic amplitude in along glider-track direction on a spatial map.

4 cm in the glider observations is entirely absent in HRET.

Here, we build upon the full-depth steric height signal of the semidiurnal coherent internal tide, which sheds light on the coherent tide along the glider track (Fig. 12b). It corresponds with the full-depth glider steric height (Fig. 12a), suggestive for prevailing tidal coherence in the glider observations. Specifically, the tidal signature at B is also present indicating at this location a potential misrepresentation of the coherent tide in HRET. This is not necessarily surprising considering that the glider sampling can access finer spatial scales. In contrast, the correct representation of the internal tide field in HRET is largely depending on the available satellite tracks for a given area that are subject to a point-wise harmonic analysis, followed by a plane-wave fit in overlapping patches. Moreover, fine-scale features may be smoothed out due to the mapping technique applied in HRET Zaron (2019).

Despite apparent deviations in the spatio-temporal variability, it is noteworthy that the modal structures, specifically modes 1-2, match closely among in-situ observations, numerical modeling, and altimetry. Among all the products, mode 1 predominates the signal accounting for $>$90 % of the baroclinic variance.

## 5 Summary and Discussion

In-situ observations from autonomous glider data have been previously exploited to infer internal-tide dynamics while providing important information on the spatial variability of high-frequency motion at fine spatial scales. Recent numerical modeling results have identified New Caledonia as a significant hot spot for internal tide generation, characterized by the westward propagation of tidal energy within well-defined tidal beams (Bendinger et al., 2023).

In this study, we inferred internal tides from a glider survey carried out in the area south of New Caledonia in the southwestern tropical Pacific over the course of $>$2 months. Spatio-temporal variability of the diurnal and semidiurnal internal tide is deduced by fitting a sinusoidal function on vertical isopycnal displacements using a least-squares method in 3-day running windows, from the surface down to 1000 m depth. The glider observations suggest a pronounced dominance of the semidiurnal tide, which can explain locally up to 80 % of the total variance of the vertical displacements. This semidiurnal dominance is in agreement with the model results in Bendinger et al. (2023). Further, our analysis reveals distinct segments of elevated tidal activity expressed by semidiurnal isopycnal displacements exceeding 20 m.

Glider observations are combined with four-dimensional regional model output. This approach serves a dual purpose. First, glider observations assess the realism of internal-tide dynamics in the numerical simulation. Second, the four-dimensional regional model outputs are used to assess the glider's capability within the upper 1000 m to infer internal tides and to retrieve their associated SSH signature. To do so, we modelled the trajectory and sampling of a pseudo glider using the hourly output

of the regional model linearly interpolated onto the glider time series with identical spatio-temporal sampling.

The observed spatio-temporal variability of the semidiurnal internal tide closely matches with the results of the pseudo glider demonstrating an overall similarity in location, magnitude, and vertical structure of the tidal signatures. The previously performed tidal analysis in Bendinger et al. (2023) suggests that these signatures sampled by both the glider and pseudo glider are associated with westward propagation of semidiurnal tidal energy, which can be traced back to the internal tide generation south/southeast of New Caledonia.

We attribute the major discrepancies between glider observations and the full-model pseudo glider to tidal incoherence induced by eddy-internal tide interactions. This is supported by a simplified ray tracing analysis which tracks the propagation of a semidiurnal ray for a given initialization region and propagation angle. In propagation direction, the semidiurnal ray may be refracted associated with variations in phase velocity induced by the background currents of mesoscale eddies affecting the tidal ray orientation. Specifically, the theoretical pathway of the refracted tidal beam intersects with the glider track, coinciding with the location and time stamp where and when departure from tidal coherence is expected in the pseudo glider.

Using glider observations alone, a single glider mission is not sufficient to distinguish between tidal coherence and incoherence. Surely, repeated glider sections would provide additional information on the coherent and incoherent variance while allowing for a model validation with more confidence. Whatsoever, we note that the complementary analysis of in-situ observations and numerical modeling can be used as a potential approach to infer where tidal incoherence may occur in glider observations. Though, this approach is limited to regions with available high-resolution model output including tidal forcing, which overlaps in space and time with glider data.

Through the vertical integration of vertical displacements, we established a connection between the semidiurnal internal tide's signature in the upper 1000 m and its expression in SSH. In the semidiurnal frequency band, the upper 1000 m in the pseudo glider simulation accounts for 78 % of the full-depth steric height variance. To encompass the entire depth range of the semidiurnal tide, extending beyond 1000m, we projected the pseudo glider's vertical displacements onto a set of climatological modes before extrapolating vertically. Utilizing a two-mode approximation, we increased the explained variance from 78 % to 93 % percent. When projecting on the first mode only, the explained variance increased to 85 %. Steric height, derived from the glider observations, shows good agreement with empirical SSH estimates from satellite altimetry, indicating the prevalence of tidal coherence during the glider mission. We point out that the methodology accounting for the full-depth range needs validation, especially in regions where high modes play a more significant role.

Linking interior dynamics of fine-scale physics with the expression in SSH has important implications for SWOT, as SWOT's SSH measurements encompass both balanced and unbalanced motions. In-situ observations play a crucial role in disentangling the physical processes in the three-dimensional water column. We have demonstrated that gliders can serve as an effective

in-situ platform for extracting the SSH signature of internal tides, particularly in the area south of New Caledonia. This region became the focus of a dedicated in-situ field campaign in boreal spring 2023, conducted within the framework of the SWOT AdAC consortium. The primary objective of this campaign was to comprehensively survey fine-scale physics, including internal tides, along two SWOT swaths during the fast-sampling phase. In an area of increased tidal activity, the SWOT observability of mesoscale and submesoscale dynamics may suffer from the dominance of unbalanced motion. For obvious reasons, the analyzed glider in this study cannot be used for the direct validation of SWOT.

Preliminary analyses using conventional satellite altimetry products have revealed the presence of multiple mesoscale eddies and frontal zones in along glider-track direction. Future work will focus on the residual signal seen in Fig. 7c and the investigation whether the glider observations corrected for the diurnal and semidiurnal tide can indeed be attributed to mesoscale and submesoscale dynamics (e.g. eddies and front), as suggested by satellite altimetry.

*Code and data availability.* This study has been conducted using EU Copernicus Marine Service Information CMEMS (https://doi.org/10.48670/moi-00148). Climatological hydrography data were obtained from CARS (http://www.marine.csiro.au/~dunn/cars2009/). The tidal analysis was performed using the COMODO-SIROCCO tools, which are developed and maintained by the SIROCCO national service (CNRS/INSU). SIROCCO is funded by INSU and Observatoire Midi-Pyrénées/Université Paul Sabatier and receives project support from CNES, SHOM, IFREMER, and ANR (https://sirocco.obs-mip.fr/other-tools/prepost-processing/comodo-tools/). The publicly available HRET products from Edward Zaron (Zaron, 2019) were downloaded from https://ingria.ceoas.oregonstate.edu/~zarone/downloads.html. The numerical model configuration (CALEDO60) used in this study is introduced and described in detail in Bendinger et al. (2023). The data to reproduce the figures can be found in Bendinger (2023a) with the associated scripts in Bendinger (2023b). The ray tracing algorithm is described in full detail in Sect. 3b in Rainville and Pinkel (2006).

*Author contributions.* AB performed the analysis and drafted the manuscript under the supervision of LG and SC. LR and CV provided the ray tracing algorithm and contributed to its analysis. GS contributed by providing preliminary code on the extraction of internal tides using glider observations. FD, FM, and J-LF were involved in the preparation of the spray glider deployments, the data collection and data postprocessing. All co-authors reviewed the manuscript and contributed to the writing and final editing.

*Competing interests.* The authors declare that they have no conflict of interest.

*Acknowledgements.* This research has been supported by the Université Toulouse III - Paul Sabatier (grant from the Ministère de l'Enseignement supérieur de la Recherche et de l'Innovation, MESRI) carried out within the PhD program of AB at the Faculty of Science and Engineering and the Doctoral School of Geosciences, Astrophysics, Space and Environmental Sciences (SDU2E). Sophie Cravatte, Lionel Gourdeau,

Fabien Durand, Frédéric Marin are funded by the Institut de Recherche pour le Développement (IRD); Luc Rainville is supported by NASA (award number 80NSSC20K1132); Clément Vic is funded by the Institut français de recherche pour l'exploitation de la mer (IFREMER); Guillaume Sérazin and Jean-Luc Fuda are funded by CNRS. This study has been partially supported through the grant EUR TESS N°ANR-18-EURE-0018 in the framework of the Programme des Investissements d'Avenir. This work is a contribution to the joint CNES-NASA project SWOT in the Tropics and is supported by the French TOSCA (la Terre, l'Océan, les Surfaces Continentales, l'Atmosphère) program and the French national program LEFE (Les Enveloppes Fluides et l'Environnement). We would like to thank the IMAGO team who were deeply involved in the preparation of the first spray glider deployments around New Caledonia from 2010 to 2013. We also thank Kyla Drushka for her invitation to host AB at the Applied Physics Laboratory in Seattle, USA, her time and fruitful discussions which significantly contributed to this study.

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
