# Peer review of "Internal tides vertical structure and steric sea surface height signature south of New Caledonia revealed by glider observations"

_EGUsphere, 2024_

## Author Comment (AC1)

**Review #1**

**Internal tides vertical structure and steric sea surface height signature south of New Caledonia revealed by glider observations**

Arne Bendinger[1,a], Sophie Cravatte[1,2], Lionel Gourdeau[1], Luc Rainville[3], Clément Vic[4], Guillaume Sérazin[4,a], Fabien Durand[1], Frédéric Marin[1], and Jean-Luc Fuda[5]

[1]Université de Toulouse, LEGOS (CNES/CNRS/IRD/UT3), Toulouse, France
[2]IRD, Centre IRD de Nouméa, New Caledonia
[3]Applied Physics Laboratory, University of Washington, Seattle, WA, USA
[4]Laboratoire d'Océanographie Physique et Spatiale, Univ. Brest, CNRS, Ifremer, IRD, IUEM, Brest, France
[5]Aix Marseille Univ., Université de Toulon, CNRS, IRD, MIO UM 110, Marseille, France
[a]now at: Laboratoire d'Océanographie Physique et Spatiale, Univ. Brest, CNRS, Ifremer, IRD, IUEM, Brest, France
[b]now at: Institut de Recherche de l'Ecole Navale (IRENav), EA 3634 - Ecole Navale, 29240, Brest, France

**Correspondence:** Arne Bendinger (arne.bendinger@univ-brest.fr)

**Summary**

*This manuscript describes glider observations of internal tides and compares them with numerical models. Generally the comparison is good and discrepancies are explained via mesoscale eddy refraction. There are two major points that need to be addressed on the CTD and near inertial waves, but otherwise the manuscript is generally clear and to the point. A nice feature*
5 *is that observations from a glider are compared to the model which is selectively sampled as the glider did the ocean.*

First of all, we would like to thank the reviewer for her/his thoughtful review on both scientific and technical aspects related to the glider observations as well as her/his useful suggestions to clarify these aspects. In the following, we would like to address the reviewer's major comments.

10 **Major comment #1**

*1. The glider CTD inlet is located on top and so upward profiles experience clean flow. Downward profiles experience distorted flow. Some justification/statistics are needed to show downward profiles are not contaminated (i.e., both raw and binned data, T-S plots of upward vs downward profiles, rms difference vs p, evidence of hysteresis, and so on).*

15 We agree with the reviewer, that it is worth checking the quality of glider downward profiles in more detail. In general, we would assume that the flow around the glider downcast profiles is disturbed by a minimum for two reasons. First, the glider wings' relatively small attack angle of 3° and the glider's quasi-horizontal motion through the water column (pitch angle of

[Figure]

**Figure R1. (a)** Mean temperature profile divided into downcast (red) and upcast (blue) and **(b)** Mean temperature offset between down- and upcast including standard deviation (shading). **(c)** and **(d)** same as **(a)** and **(b)** but for salinity.

17°). Further, a pumped CTD is in use which is expected to minimize the artifact of disturbed flow during downward profiles. Whatsoever, we followed the advice of the reviewer to double check the downward profiles. The mean temperature profile
20 for downcast and upcast are quasi identical (Fig. R1a). Further, the temperature differences between down- and upcasts are centered around zero as expected for glider dives that are randomly distributed during various phases of the tides (Fig. R1b). The large variability of the temperature differences which is maximum at mid-depths can be associated with high-frequency variability, i.e. internal tides.

25 However, the downward profiles seem to feature a small offset in salinity, which depends on depth but it is most enhanced at mid-depths ($\sim$0.04 g kg$^{-1}$ at $\sim$500 m depth), where the density gradients are strongest (Fig. R1c-d). Hysteresis is not a valid hypothesis since this would affect the temperature measurements as well. There are two possible reasons for the observed salinity differences. First, it may be linked to the cell thermal lag of the conductivity cell (e.g. Garau et al., 2011). However, as reported in Garau et al. (2011) the measurements discrepancies linked to the cell-thermal lag are $\sim$0.3 psu - an order of
30 magnitude higher than the salinity offset in our glider measurements. Second, the flow may be indeed disturbed - despite a small attack angle and a pumped CTD - as the CTD lies in the wake of the glider body.

As explained further below, the downward profiles are of importance for the sinusoidal fit given that we look at a signal of T=12 h, and it takes roughly 6 h to complete a glider dive. Thus, having twice the resolution at mid-depths is advantageous.
35 The question is to what extent the salinity offset during downward profiles affects the vertical displacement signal. In other words, is the salinity offset negligible compared to the physically-driven salinity differences between downward and upward profiles due to high-frequency variability?

We argue in the following that the vertical displacements linked to the salinity offset are small compared to the internal-tide

40  driven signal. At mid-depths (300-700 m), there is a mean salinity gradient of 1 g kg$^{-1}$, i.e. 0.025 g kg$^{-1}$ m$^{-1}$ (see Fig. R1c). In that case, the salinity offset of 0.04 g kg$^{-1}$ would correspond to a vertical displacement of less than 2 m, which is much smaller than the signal we are looking at (>10-20 m). Our glider's vertical speed is in the mean 10 cm s$^{-1}$. It is hard to imagine that a flow distortion through wake turbulence causes water entrainment over a time scale 20 s (the time needed for two vertical meters). However, even if it is the case the displacement signals linked to the maximum salinity offset as shown above are

45  much smaller than the internal-tide induced vertical displacement of isopycnal surfaces. We added a summary of the above at lines 122-131.

*For future consideration, even if the downward profiles are ok, their advantage is limited. There is no gain in resolution at the turning points; there is a factor of 2 increase in resolution at the mid point; and everywhere else it's in between. So at best the*

50  *noise is reduced by a factor of $\sqrt{2}$. Then there is the energy use for marginal gain in my opinion, which could be used instead to extend the sampling with clean, upward profiles.*

We appreciate the reviewer's future consideration to extend the sampling of clear, upward profiles. Though, the gliders are not operational anymore and no future missions are planned in this region. Considering the available glider data: even if the down-

55  ward profiles show a slight offset in salinity, we argue that the downward profiles are of great importance for the sinusoidal fit of the semidiurnal tide. Given that we look at a signal of T=12 h, and it takes roughly 6 h to complete a glider dive, having twice the resolution at mid-depths is advantageous. We refer to Rainville et al. (2013) who estimated the accuracy of the M2 and K1 amplitude and phase in a series of test waves in a Monte Carlo-like approach highlighting inter alia the importance of regular sampling at mid-depths (see their appendix A2).

60

This importance of having both down- and upward profiles also reflects in our analysis when applying the sinusoidal fit for the semidiurnal tide - particularly in regions of data gaps due to aborted glider dives or glider dives with faulty GPS data, which were excluded from the data set. In those regions, the sinusoidal fit may result in spurious signals and a low goodness of fit due to a lack of measurement points when using upward profiles only.

65  **Major comment #2**

*2.1 (a) From Fig 5 it looks like you should also fit at the inertial period  29 hrs, even though this will overlap with the diurnal signal though for the 3-day window because the frequency resolution is 1/(3 days). This raises two points. (b) In line 316, this is linked to submesoscale dynamics, which appears to ignore the inertial signal, which is also expected to be larger also in the upper ocean. Since the near-inertial displacements are similar in size to the semidiurnal displacements, the near-inertial sig-*

70  *nal will have much more kinetic energy and may then be much more energetic internal wave signal than the semi diurnal signal.*

[Figure]

**Figure R2.** Model power spectral density (PSD) of horizontal kinetic energy at 160° E, 24° S. The black dashed line indicate the peak frequencies for the M2 and K1 tides representative for the semidiurnal and diurnal frequency bands, respectively. The inertial frequency is also shown.

We thank the reviewer for her/his suggestion to fit at the inertial period even though it overlaps with the diurnal signal. However, we disagree that the near-inertial displacements are similar in size to the semidiurnal displacements. We will support this in the following by the theoretical estimation of the semidiurnal and near-inertial vertical displacements at a given latitude with an a priori knowledge of their kinetic energy (Gill, 1982). The ratio of kinetic energy (KE) to potential energy (PE) for inertial gravity waves (IGWs) is given by equation 8.3.5 in Gill (1982):

$$\frac{KE}{PE} = \frac{\omega^2 + f^2}{\omega^2 - f^2} \tag{1}$$

PE is proportional to $\eta^2 N^2$, i.e. for semidiurnal tides $PE \propto \eta_{SD}^2 N^2$ and near-inertial waves $PE \propto \eta_{NI}^2 N^2$. With equation 8.3.5, this allows for an estimate of the ratio of vertical displacements between near-inertial waves and semidiurnal tides:

$$\frac{\eta_{NI}}{\eta_{SD}} = \left( \frac{KE_{NI}}{KE_{SD}} \ \frac{\omega_{NI}^2 - f^2}{\omega_{NI}^2 + f^2} \ \frac{\omega_{SD}^2 + f^2}{\omega_{SD}^2 - f^2} \right)^{1/2} \tag{2}$$

The ratio $\frac{KE_{NI}}{KE_{SD}} \leq 10^{-1}$ is estimated by the HKE power spectrum in Fig. R2. With $f = 5.92 \cdot 10^{-5} \ \mathrm{s}^{-1}$, $\omega_{NI} = 1.05 - 1.15f$, and $\omega_{SD} = 1.405 \cdot 10^{-4} \ \mathrm{s}^{-1}$, this gives a range of $\frac{\eta_{NI}}{\eta_{SD}} = 0.08 - 0.14$. This corresponds to a range of $\eta_{NI} = 1.2 - 2.1$ m, which is not only smaller than the semidiurnal displacements ($\eta_{SD} = 15$ m) but also smaller than the observed residual (see Fig. R3a

[Figure]

**Figure R3.** Model vertical displacements at $60°$ E, $24°$ S during the glider deployment period August-October 2014 for the **(a)** subinertial, **(b)** semidiurnal, **(c)** diurnal, and **(d)** near-inertial frequency band. The wind stress is also given in **(d)**.

or Fig. 5 in the preprint manuscript).

85

Supported by theory, there is no reason for us to believe that the residual is dominated by near-inertial motion. This becomes even clearer when considering that diurnal and near-inertial signals partly overlap in the 3-day window fits and, hence, the residual should be linked to anything which is not semidiurnal, not diurnal, and to a large extent not near-inertial.

90   To further support our point of view, we filtered numerical simulation output for a fixed point in the semidiurnal, diurnal, near-inertial, and subinertial frequency band. The semidiurnal and diurnal frequency band was determined by a bandpass filter with a bandwidth $[c^{-1}\omega, c\omega]$, where $c = 1.1$. $c$ was chosen such that the diurnal frequency band does not overlap with the in- ertial frequency. The subinertial frequency band was defined as $< 1.1f$. The subinertial, semidiurnal, diurnal, and near-inertial vertical displacements at $160°$ E, $24°$ S during the glider deployment period August-October 2014 are shown in Fig. R3. At

95   superinertial frequencies, it is clearly the semidiurnal tide which dominates the vertical displacements (Fig. R3b). As expected from the power spectrum in Fig. R2, the diurnal signal is significantly reduced in amplitude (Fig. R3c). The same applies for the near-inertial signal.

[Figure]

**Figure R4.** (a) Glider observations derived vertical displacements (black) for an exemplary 3-day window at 300 m depth and the sinusoidal fit of the diurnal (blue), semidiurnal (red) internal tide, and the residual signal (gray). The respective explained variability is also shown. (b) Same as (a) but additionally fit at the inertial period.

In reality, the diurnal and near-inertial signals overlap in the 3-day window fit as pointed out by the reviewer. Figure R4 shows
100 an exemplary 3-day window when fitting for the semidiurnal and diurnal peak frequency only (Fig. R4a) and when including the inertial peak frequency (Fig. R4b). Though, fitting at the inertial period in addition to the semidiurnal and diurnal period raises concerns regarding its physical interpretation.

We added parts of above justification in the following paragraphs lines 207-210 and lines 339-350.
105

*2.2 (c) Any comparison with the model should be a comparison of semi diurnal and not individual constituents, ie M2, S2, which it seems to me is done most/all of the time.*

We agree that the comparison with the model should be a comparison of the semidiurnal components. For clarification, the
110 glider observations are indeed compared with the model's semidiurnal signal (reconstructed from the M2, S2, N2 signals). The reviewer might have misunderstood this. We refer to lines 196-199.

**Minor comments**

*3. It would be worth mentioning how the internal tides near New Caledonia show up in previous global models and observations by Zhao, Niwa and Hibiya, Ray and Cartwright, etc. and how your models and observations compare.*

115

We thank the reviewer for her/his suggestion to consider how internal tides near New Caledonia are represented in global models and observations. In this context, we refer to Bendinger et al. (2023) who validated not only the model's capability to realistically reproduce motion from the large-scale circulation, down to high-frequency motion while justifying the model's eligibility to simulate internal-tide dynamics. Here, we will briefly recall the kinetic energy levels and the internal tide's SSH

120 signature.

[Figure]

**Figure R5.** Power spectral density of near surface (20-100 m) horizontal kinetic energy for CALEDO60 without (blue) and with (red) tidal forcing for the full model time series near 167.25° E, 20.43° S in the New Caledonian eastern boundary current (see Fig. 2 in Bendinger et al., 2023). The energy spectra are compared to a mooring time series that was deployed between November 2010 and October 2011 (Durand et al., 2017). The vertical dashed black lines are representative of the inertial frequency $f$, the peak frequency of the K1 diurnal tide, and the peak frequency of the M2 semidiurnal tide.

Kinetic energy levels were validated against moored in-situ observations revealing that inertial and tidal energy peaks are very close to the observations (Fig. R5). Furthermore, for higher frequencies ($> f$), the simulation with tidal forcing (red line) introduces a major improvement to the simulation without tidal forcing (blue line).

125 The realism of the governing M2 internal tide was investigated by the internal tide's expression in SSH while being compared to the satellite-altimetry-derived empirical estimates from the HRET model (Zaron, 2019). The M2 SSH amplitude for modes

1-2 is shown in Fig. R6. It is in good agreement with the HRET model concerning the spatial representation of the M2 SSH for both mode 1 and mode 2. Overall, mode 1 seems to be enhanced in our model, whereas mode 2 is underestimated in some regions. Note that the given differences may be associated with the different time periods the datasets are referenced to as well as the length of the time series for the model (1 year) and altimetry (25 years). We added a small paragraph on the model's validation in Sect. 2.2, lines 148-154.

We did not explicitly look at the global models from Niwa and Hibiya (2001, 2014); Shriver et al. (2012); Buijsman et al. (2017) and how they compare to our simulation, but we are aware of the mentioned studies. Even though it is out of scope for this study, we had a brief look in the MITGCM whose representation of internal tides around New Caledonia is very limited. This is primarily linked to the rather course bathymetry product which does not resolve the fine-scale bathymetry in the New Caledonia internal tide hot-spot regions. This is why our model's bathymetry has received very careful attention. The bathymetry is composed of the GEBCO_2019 grid and a compilation of multibeam echosounder data acquired over the years in the New Caledonia economic zone (see Sect. 2.1 in Bendinger et al., 2023).

*4. For interference patterns there is a paper by Rainville et al about Hawaii.*

We kindly ask the reviewer to clarify this comments/suggestion. We are aware of the mentioned study from Rainville et al. (2010). However, in our case there is a very well defined tidal energy source (Pines Ridge) and the glider survey was carried in the vicinity to it. We argue that the refractive impact of the mesoscale eddies is most relevant. We do not discuss interference patterns from multiple sources. This would be more relevant for farfield propagation.

**Comments by line**

*44 - The smearing of temporal variability into spatial variability by slowly moving gliders has been examined by Rudnick & Cole (2011). You can extract some wavelengths based on your dominant diurnal and semidiurnal periods.*

We thank the reviewer for this comment. First of all, we added the reference for Rudnick and Cole (2011) and Rainville et al. (2013) since these studies tackle and investigate the extraction of low- and high-frequency signals from a moving glider platform (lines 55-57). Further, we computed the horizontal wavenumber spectrum of conservative temperature at constant depth (300 m, see Fig. R7). According to Rudnick and Cole (2011), the glider observations underlie Doppler smearing (due to glider's finite speed and aliasing due to discrete sampling), which in turn is responsible for the projection from higher wavenumbers onto lower wavenumbers. A break in slope would indicate the minimum resolved wavenumber. Similarly to Rudnick and Cole (2011), we find the break of slope around 30 km wavelength.

[Figure]

**Figure R6.** CALEDO60 M2 SSH amplitude for **(a)** mode 1 and **(b)** mode 2 in comparison with the empirical estimates of the High Resolution Empirical Tide (HRET) model for **(c)** mode 1 and **(d)** mode 2.

*48 - Could lead with the advantages of using gliders for studying internal waves and then follow with limitations (line 37).*

160

We thank the reviewer for this comments. We applied her/his suggestion to start with the glider's advantages followed by their limitations in lines 46-57.

*58 - This paragraph could benefit from the addition of some of the original references related to Hawaii, theory, altimetry,*
165 *interference, and coherence.*

We agree with the reviewer and added some reference in the paragraph (lines 59-77).

*127 - Some more explanation is needed of how forcing by tidal potential and at the boundaries works. Usually only one type*
170 *of forcing is used because the 2 methods may provide conflicting information, which would produce spurious waves at the*

*boundaries and perhaps elsewhere. Or perhaps only at certain times when the mismatches are more apparent.*

We appreciate the reviewer's comment on the model's tidal forcing. We recall in the following the model's set up while refer-
ring to Bendinger et al. (2023). For clarification, the model's set up consists of a host grid (TROPICO12) which covers most of
175  the subtropical to tropical Pacific Ocean and a nesting grid (CALEDO60) in the southwestern Pacific encompassing New Cale-
donia. A two-way nesting framework (AGRIF) ensures the lateral boundary coupling between TROPICO12 and CALEDO60.
As stated in the preprint, both domains are forced by the astronomical tidal potential of the five major tidal constituents (M2,
S2, N2, K1, O1). TROPICO12 is additionally forced at its open lateral boundaries by barotropic SSH and currents (FES2014,
Lyard et al., 2021).

180

We were surprised about the reviewer's comments usually only one type of forcing is used, i.e. either tidal forcing at the lateral
boundaries or tidal potential in the domains' interior. To our knowledge, this is common practice such as in the recent studies
from Ruault et al. (2020) and Barbot et al. (2022). The reviewer stated that applying both methods may provide conflicting

[Figure]

**Figure R7.** Power spectrum density of conservative temperature at 300 m depth. The dashed gray line indicates a $k^{-2}$ slope for reference.

information, which would produce spurious waves at the boundaries and perhaps elsewhere. This is not what we observe as shown in Fig. R8, neither at the lateral boundaries of TROPICO12, nor those of CALEDO60, in which the M2 SSH amplitude (shading) and phase (contour) based on a 1-year (2014) harmonic analysis (assuming that model SSH is dominated by the barotropic tide) in comparison to the global tide atlas FES2014.

Further, we suggest that in fact both the tidal forcing of barotropic SSH and currents at the lateral boundaries of TROPICO12 in addition to the astronomical potential in the domain's interior are of importance. In larger domains like TROPICO12, where the domain size significantly exceeds the barotropic tide length scale, incorporating both tidal potential in the interior and tidal forcing at lateral boundaries appears to be crucial for accurately representing the tidal dynamics. In contrast, if the domain's size is small, i.e. smaller than the barotropic length scale, tidal forcing at the lateral boundaries is sufficient. Forcing by the astronomical tide potential alone is, however, not sufficient as this would create discontinuities along the lateral boundaries.

*148 - Absolute geostrophic currents referenced to glider depth mean velocity can be calculated too.*

[Figure]

**Figure R8.** M2 SSH amplitude (shading) and phase (contour) for (a) TROPICO12 and (c) CALEDO60 based on a 1-year (2014) harmonic analysis (assuming that the model SSH is dominated by the barotropic tide) and in comparison to the global tide atlas FES2014 for (b) TROPICO12 and (d) CALEDO60. The boxes in (a) and (b) indicate the location of the CALEDO60 domain.

We appreciate the reviewer's suggestion. However, to understand the origin and fate of the internal tide - from the generation site to the location along the glider track where it was sampled - it is crucial to have full information of the 2-dimensional velocity field. Therefore, we rely on two-dimensional gridded satellite altimetry data.

*168 - Also smearing of the semidiurnal signal goes into scales < 30 km (Rudnick & Cole, 2011) and so averaging should be done over larger scales.*

This paragraph concerns the justification of the sinusoidal fit applied in running 3-day windows. If we understand correctly, the reviewer suggests that the vertical displacements should be computed relative to a time period longer than the 3-day window. We followsthe methodology described in Rainville et al. (2013). We also tested longer windows, but noticed that this primarily results into smoothing of the fitted amplitude and phase.

*265 - This isn't a big deal because it's the time differences (ie, 6 hrs) between profiles that is the main thing.*

We agree with the reviewer that it is the time difference of 6 hours between profiles which is driving factor. However, our analysis suggests that interpolating the vertical displacements onto a common time stamp may slightly increase the explained variability for $h_{\mathrm{pseudo}}^{1000\mathrm{m}}=0.78$, $h_{\mathrm{pseudo}}^{\mathrm{full}}\Sigma_{n=1}^{1}=0.85$, $h_{\mathrm{pseudo}}^{\mathrm{full}}\Sigma_{n=1}^{2}=0.93$ from Table 1 - compared to $h_{\mathrm{pseudo}}^{1000\mathrm{m}}=0.67$, $h_{\mathrm{pseudo}}^{\mathrm{full}}\Sigma_{n=1}^{1}=0.74$, $h_{\mathrm{pseudo}}^{\mathrm{full}}\Sigma_{n=1}^{2}=0.81$, when not interpolating onto a common time stamp.

*275 - It would be helpful to see the full depth structure of modes 1-2.*

Following the reviewer's suggestion, we show in the corrected manuscript in Fig. 11 now in addition to the pseudo glider vertical displacements in the upper 1000 m (Fig. 11b), also the pseudo glider vertical displacements extrapolated to full-depth using projections on climatological modes 1-2 (Fig. 11c).

*Fig 7- no advantage to using grey shading*

We thank the reviewer for this remark. This was also noted by reviewer #2. We changed the colormap in Fig. 7.

*Fig 11 - there are some spurious signals in the model or interpolating it onto the glider path especially near 15/8, 8/10 and 16/10.*

These spurious signals are not physical but linked to the glider data visualization in regions with data gaps. These data gaps are linked to aborted glider dives or glider dives with faulty GPS data which is indeed the case for the time stamps near 15/8, 8/10 and 16/10 as stated by the reviewer. Since the data is not gridded along the time axis, these data gaps may appear as spurious

signals.

*Fig 11 - explain why the upper 100 m is missing.*

Indeed no explication is given in the manuscript. We thank the reviewer for this remark. We added the following sentence in lines 434-436: *Note that we exclude in the following analysis the near-surface ocean layer, i.e the upper 100 m, which might be contaminated by mixed layer and mesoscale to submesoscale dynamics expressed by the elevated residual in Fig. 7c.*

*310 - semi diurnal displacement variance. See comment 2 about total internal wave signal.*

We made slight modifications in line 331-332: *Locally, the semidiurnal displacement variance explains up to 80 % of the total displacement variance (Fig. 7b)*

**References**

Barbot, S., Lagarde, M., Lyard, F., Marsaleix, P., Lherminier, P., and Jeandel, C.: Internal tides responsible for lithogenic inputs along the Iberian continental slope, Journal of Geophysical Research: Oceans, 127, e2022JC018 816, https://doi.org/10.1029/2022JC018816, 2022.

Bendinger, A., Cravatte, S., Gourdeau, L., Brodeau, L., Albert, A., Tchilibou, M., Lyard, F., and Vic, C.: Regional modeling of internal-tide dynamics around New Caledonia – Part 1: Coherent internal-tide characteristics and sea surface height signature, Ocean Science, 19, 1315–1338, https://doi.org/10.5194/os-19-1315-2023, 2023.

Buijsman, M. C., Arbic, B. K., Richman, J. G., Shriver, J. F., Wallcraft, A. J., and Zamudio, L.: Semidiurnal internal tide incoherence in the equatorial Pacific, Journal of Geophysical Research: Oceans, 122, 5286–5305, https://doi.org/10.1002/2016JC012590, 2017.

Durand, F., Marin, F., Fuda, J.-L., and Terre, T.: The east caledonian current: a case example for the intercomparison between altika and in situ measurements in a boundary current, Marine Geodesy, 40, 1–22, https://doi.org/10.1080/01490419.2016.1258375, 2017.

Garau, B., Ruiz, S., Zhang, W. G., Pascual, A., Heslop, E., Kerfoot, J., and Tintoré, J.: Thermal lag correction on Slocum CTD glider data, Journal of Atmospheric and Oceanic Technology, 28, 1065–1071, https://doi.org/10.1175/JTECH-D-10-05030.1, 2011.

Gill, A. E.: Atmosphere-ocean dynamics, vol. 30, Academic press, 1982.

Lyard, F. H., Allain, D. J., Cancet, M., Carrère, L., and Picot, N.: FES2014 global ocean tide atlas: design and performance, Ocean Science, 17, 615–649, https://doi.org/10.5194/os-17-615-2021, 2021.

Niwa, Y. and Hibiya, T.: Numerical study of the spatial distribution of the M2 internal tide in the Pacific Ocean, Journal of Geophysical Research: Oceans, 106, 22 441–22 449, https://doi.org/10.1029/2000JC000770, 2001.

Niwa, Y. and Hibiya, T.: Generation of baroclinic tide energy in a global three-dimensional numerical model with different spatial grid resolutions, Ocean Modelling, 80, 59–73, https://doi.org/10.1016/j.ocemod.2014.05.003, 2014.

Rainville, L., Johnston, T. S., Carter, G. S., Merrifield, M. A., Pinkel, R., Worcester, P. F., and Dushaw, B. D.: Interference pattern and propagation of the M2 internal tide south of the Hawaiian Ridge, Journal of physical oceanography, 40, 311–325, https://doi.org/10.1175/2009JPO4256.1, 2010.

Rainville, L., Lee, C. M., Rudnick, D. L., and Yang, K.-C.: Propagation of internal tides generated near Luzon Strait: Observations from autonomous gliders, Journal of Geophysical Research: Oceans, 118, 4125–4138, https://doi.org/10.1002/jgrc.20293, 2013.

Ruault, V., Jouanno, J., Durand, F., Chanut, J., and Benshila, R.: Role of the tide on the structure of the Amazon plume: A numerical modeling approach, Journal of Geophysical Research: Oceans, 125, e2019JC015 495, https://doi.org/10.1029/2019JC015495, 2020.

Rudnick, D. L. and Cole, S. T.: On sampling the ocean using underwater gliders, Journal of Geophysical Research: Oceans, 116, https://doi.org/10.1029/2010JC006849, 2011.

Shriver, J., Arbic, B. K., Richman, J., Ray, R., Metzger, E., Wallcraft, A., and Timko, P.: An evaluation of the barotropic and internal tides in a high-resolution global ocean circulation model, Journal of Geophysical Research: Oceans, 117, https://doi.org/10.1029/2012JC008170, 2012.

Zaron, E. D.: Baroclinic tidal sea level from exact-repeat mission altimetry, Journal of Physical Oceanography, 49, 193–210, https://doi.org/10.1175/JPO-D-18-0127.1, 2019.

---

## Author Comment (AC2)

**Review #2**

**Internal tides vertical structure and steric sea surface height signature south of New Caledonia revealed by glider observations**

Arne Bendinger[1,a], Sophie Cravatte[1,2], Lionel Gourdeau[1], Luc Rainville[3], Clément Vic[4], Guillaume Sérazin[4,a], Fabien Durand[1], Frédéric Marin[1], and Jean-Luc Fuda[5]

[1]Université de Toulouse, LEGOS (CNES/CNRS/IRD/UT3), Toulouse, France
[2]IRD, Centre IRD de Nouméa, New Caledonia
[3]Applied Physics Laboratory, University of Washington, Seattle, WA, USA
[4]Laboratoire d'Océanographie Physique et Spatiale, Univ. Brest, CNRS, Ifremer, IRD, IUEM, Brest, France
[5]Aix Marseille Univ., Université de Toulon, CNRS, IRD, MIO UM 110, Marseille, France
[a]now at: Laboratoire d'Océanographie Physique et Spatiale, Univ. Brest, CNRS, Ifremer, IRD, IUEM, Brest, France
[b]now at: Institut de Recherche de l'Ecole Navale (IRENav), EA 3634 - Ecole Navale, 29240, Brest, France

**Correspondence:** Arne Bendinger (arne.bendinger@univ-brest.fr)

**Summary**

*This paper describes an effort to investigate internal tide structure using glider observations. I found this paper to be interesting and not hard to follow.*

5 We thank the reviewer for her/his review. In the following, we would like to address the reviewer's minor comments. The reviewer stated that she/he had additional thoughts which were enumerated by reviewer #1. We therefore refer to review #1 for additional information.

**Minor comments**

*Line 8-9 – eddy-internal tide interactions are certainly one source of discrepancy. I don't discount this mechanism, but there*
10 *are other potential sources of error as well (such as topography and stratification). What about those?*

We agree with the reviewer that other mechanisms and sources of errors exist (e.g. topography and stratification). Eddy-internal tide interactions as a major driver for the discrepancies relies on the findings deduced from the ray tracing. In contrast to the current's impacts, we find that the impact of topography and stratification is of less importance on the spatial scales that we
15 consider here in internal tide propagation direction (compare \wo currents with \w currents scenario in Fig. 9). For the sake of simplicity, we applied the ray tracing on bathymetry from ETOPO2v2 (Smith and Sandwell, 1997) and climatological stratification from the World Ocean Atlas (Locarnini et al., 2018; Zweng et al., 2019). For evident reasons, the latter does not take

into account eddy-induced stratification changes in internal tide propagation direction - a potential source for tidal incoherence. Though, it has recently been shown in a more realistic ray tracing approach that the eddy-associated currents make a greater contribution to internal tide refraction than eddy-associated stratification (Guo et al., 2023). We made modifications by replacing *primarily* by *in large part* in lines 8-9.

*Line 14 – "glider observations' predominating coherent nature". Why would this be the case? I assume there isn't something intrinsic about the glider observations that would make them have a coherent nature. Is the observation record too short to observe much incoherence, was this a time period of low variability or some other reason?*

We agree with the reviewer that the phrase *glider observations' predominating coherent nature* may be confusing. We changed it in lines 12-14: *Notably, the steric SSH from glider observations aligns closely with empirical estimates derived from satellite altimetry, highlighting the internal tide's predominant coherent nature during the glider's sampling.*

*Lines 44-46 – problems separating high and low frequency signals. Can these be separated enough to get insight into the coherent/incoherent question?*

[Figure]

**Figure R1.** Power spectrum density of conservative temperature at 300 m depth. The dashed gray line indicates a $k^{-2}$ slope for reference.

This is clearly a fair question to ask. Despite the limitations that are associated with the glider sampling, we believe that we

35  were able to show that the glider observations were successfully exploited to extract the high-frequency internal-tide signal. The coherent semidiurnal signal from the model (analysed on the model grid and not using the pseudo-glider sampled data) gives us confidence in this regard (compare Fig. 8a and Fig. 8c). Further, the conclusions made on tidal incoherence are supported by the ray tracing to show that the potential departure from tidal coherence (compare Fig. 8b and Fig.8c) can be indeed attributed to eddy-internal tide interactions through tidal beam refraction (Fig. 9-10).

40

In the new manuscript we refer to Rudnick and Cole (2011) and Rainville et al. (2013) who address this question with regard to the glider sampling (lines 55-57). According to Rudnick and Cole (2011), the glider observations underlie Doppler smearing (due to glider's finite speed and aliasing due to discrete sampling), which in turn is responsible for the projection from higher wavenumbers onto lower wavenumbers. A break in slope would indicate the minimum resolved wavenumber. Similarly to

45  Rudnick and Cole (2011), we find the break of slope around 30 km wavelength (see Fig. R1)

*Line 121 - the numerical model used, has it been compared against other models and/or observations to show it is sufficiently accurate/realistic to aid in this study?*

50  We kindly refer to Bendinger et al. (2023) in which the numerical model has been introduced and largely validated. Further, it contains an assessment on the model's capability to realistically reproduce motion from the large-scale circulation down to high-frequency motion while justifying the model's eligibility to simulate internal-tide dynamics. Here, we will recall the main conclusions.

55  The mean circulation was assessed using the Argo-CARS merged velocity product from Kessler and Cravatte (2013) revealing a good representation of the regional circulation. The model accurately depicts the westward zonal jets with well-located positions and reasonable amplitudes (see Fig. 2 in Bendinger et al., 2023. Mesoscale eddy variability was validated against satellite altimetry. Overall, the spatial pattern of simulated EKE is in good agreement (see Fig. 3 in Bendinger et al., 2023). Maximum levels of EKE are elevated in the model south of New Caledonia. Though, this can be inter alia attributed to the conventional

60  two-dimensional gridded satellite altimetry products, which do not resolve wavelengths smaller than 150-200 km in our region and may miss the contribution of smaller-scale dynamics.

Kinetic energy levels were validated against moored in-situ observations revealing that model energy levels are very close to observations from seasonal to inertial timescales (180 d to 36 h), i.e., for mesoscale and submesoscale processes. Inertial and

65  tidal energy peaks are also in good agreement (Fig. R2. Furthermore, for higher frequencies ($> f$), the simulation with tidal forcing (red line) introduces a major improvement to the simulation without tidal forcing (blue line). This validation, even if only performed at one location, gives us confidence in the ability of the numerical simulation to correctly represent the tides

and their interaction with mesoscale processes.

70  The realism of the governing M2 internal tide was investigated by the internal tide's expression in SSH while being compared to the satellite-altimetry-derived empirical estimates from the HRET model (Zaron, 2019). The M2 SSH amplitude for modes 1-2 is shown in Fig. R3. It is in good agreement with the HRET model concerning the spatial representation of the M2 SSH for both mode 1 and mode 2. Overall, mode 1 seems to be enhanced in our model, whereas mode 2 is underestimated in some regions. Note that the given differences may be associated with the different time periods the datasets are referenced to as well

75  as the length of the time series for the model (1 year) and altimetry (25 years).

We added a small paragraph on the model's validation in Sect. 2.2, lines 148-154.

*Line 127 – tidal forcing along boundary. It appears you're forcing with FES along the boundary. This is barotropic forcing*
80  *only. Mazloff et al (2020-JGR) talk about the importance of remote forcing for regional modeling of internal waves. Can you comment on the results of this paper and either it's relevance or lack of relevance to this work?*

[Figure]

**Figure R2.** Power spectral density of near surface (20-100 m) horizontal kinetic energy for CALEDO60 without (blue) and with (red) tidal forcing for the full model time series near 167.25° E, 20.43° S in the New Caledonian eastern boundary current (see Fig. 2 in Bendinger et al., 2023). The energy spectra are compared to a mooring time series that was deployed between November 2010 and October 2011 (Durand et al., 2017). The vertical dashed black lines are representative of the inertial frequency $f$, the peak frequency of the K1 diurnal tide, and the peak frequency of the M2 semidiurnal tide.

We appreciate the reviewer's comments. It is indeed relevant for this study. We are aware of the importance of remote forcing for regional modeling of internal waves (e.g. Nelson et al., 2020; Mazloff et al., 2020; Siyanbola et al., 2023). In our model configuration, we account for high-frequency oceanic variability from remote regions by employing a sufficiently large host-grid domain (TROPICO12). Within this domain, barotropic tide forcing is applied at the open lateral boundaries, and a two-way lateral boundary coupling is maintained between the host (TROPICO12) and nesting grid (CALEDO60) throughout the simulation. The latter ensures the free internal wave propagation from remote regions into the regional domain. The model's domain configuration is shown in Fig. R4. For more information, we kindly refer to Sect. 2.1 in Bendinger et al. (2023).

*Figure 7 – I find this gray scale plot hard to interpret. The other plots in the paper are color, was there a reason why gray scale was used? I may be missing something that this color bar choice was intended to help convey.*

We thank the reviewer for this remark. This was also noted by reviewer #1. We changed the colormap in Fig. 7.

[Figure]

**Figure R3.** CALEDO60 M2 SSH amplitude for **(a)** mode 1 and **(b)** mode 2 in comparison with the empirical estimates of the High Resolution Empirical Tide (HRET) model for **(c)** mode 1 and **(d)** mode 2.

*Line 255 – "have never been used the SSH signature" doesn't read right. Did you intend to convey "have never been used to derive the SSH signature"?*

We thank the reviewer for spotting this little mistake. Indeed, we meant *have never been used to derive the SSH signature*.

*Line 344 – There are other sources of potential disagreement as well, such as forcing, topography and stratification errors and unconstrained variability. Have you explored or considered those?*

We acknowledge the sources of potential disagreement stated by the reviewer. The stratification has been validated against climatology in Bendinger et al. (2023) revealing a good agreement of the normalized modal structures for the four lowest modes and for both the displacement and vertical velocity (see their Fig. 6). The topography has also received very careful attention. The bathymetry is composed of the GEBCO_2019 grid and a compilation of multibeam echosounder data acquired over the years in the New Caledonia economic zone (see Sect. 2.1 in Bendinger et al., 2023). In this way, we ensure the accurate representation of fine-scale bathymetric features, including ridges and seamounts around New Caledonia. Figure 8a-c together with the ray tracing results in Fig. 9-10 strongly suggest that the main source of discrepancy is tidal incoherence. Nonetheless, other error sources certainly exist. We replaced the paragraph in lines 340-344 to the following: *Given that both glider observations and the full-model pseudo glider feature identical sampling, variations linked to the spring-neap tide cycle are not valid hypotheses. Potential sources for the discrepancies may lie in the erroneous representation of the model's*

[Figure]

**Figure R4.** Model setup showing the host grid domain (TROPICO12, yellow box) and the nesting grid (CALEDO60, white box) including the bathymetry (shading) and the SWOT CalVal orbit (black transparent lines) with the highlighted ground track (red line) that crosses the CALEDO60 domain.

*bathymetry and/or stratification leading to inaccuracies in simulating the precise beam location or the model's vertical mode*
115 *structure. Though, the used bathymetry product has received careful attention in the model configuration and is believed to accurately represent fine-scale bathymetric features while stratification was validated against climatology (Bendinger et al., 2023)* (lines 375-380 in the new manuscript).

**References**

Bendinger, A., Cravatte, S., Gourdeau, L., Brodeau, L., Albert, A., Tchilibou, M., Lyard, F., and Vic, C.: Regional modeling of internal-tide dynamics around New Caledonia – Part 1: Coherent internal-tide characteristics and sea surface height signature, Ocean Science, 19, 1315–1338, https://doi.org/10.5194/os-19-1315-2023, 2023.

Durand, F., Marin, F., Fuda, J.-L., and Terre, T.: The east caledonian current: a case example for the intercomparison between altika and in situ measurements in a boundary current, Marine Geodesy, 40, 1–22, https://doi.org/10.1080/01490419.2016.1258375, 2017.

Guo, Z., Wang, S., Cao, A., Xie, J., Song, J., and Guo, X.: Refraction of the M2 internal tides by mesoscale eddies in the South China Sea, Deep Sea Research Part I: Oceanographic Research Papers, 192, 103 946, https://doi.org/10.1016/j.dsr.2022.103946, 2023.

Kessler, W. S. and Cravatte, S.: Mean circulation of the Coral Sea, Journal of Geophysical Research: Oceans, 118, 6385–6410, https://doi.org/10.1002/2013JC009117, 2013.

Locarnini, M., Mishonov, A., Baranova, O., Boyer, T., Zweng, M., Garcia, H., Seidov, D., Weathers, K., Paver, C., and Smolyar, I.: World ocean atlas 2018, volume 1: Temperature, NOAA Atlas NESDIS 81, p. 52pp, https://archimer.ifremer.fr/doc/00651/76338/, 2018.

Mazloff, M. R., Cornuelle, B., Gille, S. T., and Wang, J.: The importance of remote forcing for regional modeling of internal waves, Journal of Geophysical Research: Oceans, 125, e2019JC015 623, https://doi.org/10.1029/2019JC015623, 2020.

Nelson, A., Arbic, B., Menemenlis, D., Peltier, W., Alford, M., Grisouard, N., and Klymak, J.: Improved internal wave spectral continuum in a regional ocean model, Journal of Geophysical Research: Oceans, 125, e2019JC015 974, https://doi.org/10.1029/2019JC015974, 2020.

Rainville, L., Lee, C. M., Rudnick, D. L., and Yang, K.-C.: Propagation of internal tides generated near Luzon Strait: Observations from autonomous gliders, Journal of Geophysical Research: Oceans, 118, 4125–4138, https://doi.org/10.1002/jgrc.20293, 2013.

Rudnick, D. L. and Cole, S. T.: On sampling the ocean using underwater gliders, Journal of Geophysical Research: Oceans, 116, https://doi.org/10.1029/2010JC006849, 2011.

Siyanbola, O. Q., Buijsman, M. C., Delpech, A., Renault, L., Barkan, R., Shriver, J. F., Arbic, B. K., and McWilliams, J. C.: Remote internal wave forcing of regional ocean simulations near the US West Coast, Ocean Modelling, 181, 102 154, https://doi.org/10.1016/j.ocemod.2022.102154, 2023.

Smith, W. H. and Sandwell, D. T.: Global sea floor topography from satellite altimetry and ship depth soundings, Science, 277, 1956–1962, https://doi.org/10.1126/science.277.5334.1956, 1997.

Zaron, E. D.: Baroclinic tidal sea level from exact-repeat mission altimetry, Journal of Physical Oceanography, 49, 193–210, https://doi.org/10.1175/JPO-D-18-0127.1, 2019.

Zweng, M., Seidov, D., Boyer, T., Locarnini, M., Garcia, H., Mishonov, A., Baranova, O., Weathers, K., Paver, C., Smolyar, I., et al.: World ocean atlas 2018, volume 2: Salinity, NOAA Atlas NESDIS 82, p. 50pp, https://archimer.ifremer.fr/doc/00651/76339/, 2019.